# RNaseH2A downregulation drives inflammatory gene expression via genomic DNA fragmentation in senescent and cancer cells

Sho Sugawara [1,16], Ryo Okada[1,2,16], Tze Mun Loo[1], Hisamichi Tanaka[1], Kenichi Miyata[1], Masatomo Chiba[1], Hiroko Kawasaki[1], Kaoru Katoh[3], Shizuo Kaji [4], Yoshiro Maezawa[5], Koutaro Yokote[5], Mizuho Nakayama[6], Masanobu Oshima [6], Koji Nagao[7], Chikashi Obuse[7], Satoshi Nagayama [8,9], Keiyo Takubo [10], Akira Nakanishi [2], Masato T. Kanemaki [11,12], Eiji Hara [13] & Akiko Takahashi [1,14,15✉]

Cellular senescence caused by oncogenic stimuli is associated with the development of various age-related pathologies through the senescence-associated secretory phenotype (SASP). SASP is mediated by the activation of cytoplasmic nucleic acid sensors. However, the molecular mechanism underlying the accumulation of nucleotide ligands in senescent cells is unclear. In this study, we revealed that the expression of RNaseH2A, which removes ribonucleoside monophosphates (rNMPs) from the genome, is regulated by E2F transcription factors, and it decreases during cellular senescence. Residual rNMPs cause genomic DNA fragmentation and aberrant activation of cytoplasmic nucleic acid sensors, thereby provoking subsequent SASP factor gene expression in senescent cells. In addition, RNaseH2A expression was significantly decreased in aged mouse tissues and cells from individuals with Werner syndrome. Furthermore, RNaseH2A degradation using the auxin-inducible degron system induced the accumulation of nucleotide ligands and induction of certain tumourigenic SASP-like factors, promoting the metastatic properties of colorectal cancer cells. Our results indicate that RNaseH2A downregulation provokes SASP through nucleotide ligand accumulation, which likely contributes to the pathological features of senescent, progeroid, and cancer cells.

[1] Division of Cellular Senescence, Cancer Institute, Japanese Foundation for Cancer Research, Koto-ku, Tokyo 135-8550, Japan. [2] Department of Molecular Genetics, Medical Research Institute, Tokyo Medical and Dental University (TMDU), Bunkyo-ku, Tokyo 113-8510, Japan. [3] Biomedical Research Institute, National Institute of Advanced Industrial Science and Technology (AIST), Tsukuba, Ibaraki 305-8560, Japan. [4] Institute of Mathematics for Industry, Kyushu University, Nishi-ku, Fukuoka 819-0395, Japan. [5] Department of Endocrinology, Hematology and Gerontology, Graduate School of Medicine, Chiba University, Chiba, Chiba 260-0856, Japan. [6] Division of Genetics, Cancer Research Institute, Kanazawa University, Kanazawa 920-1192, Japan. [7] Laboratory of Genome Structure and Function, Graduated School of Science, Osaka University, Toyonaka, Osaka 560-0043, Japan. [8] Department of Gastroenterological Surgery, Cancer Institute Hospital, Japanese Foundation for Cancer Research, 135-8550 Tokyo, Japan. [9] Department of Surgery, Uji-Tokushukai Medical Center, Kyoto 611-0041, Japan. [10] Department of Stem Cell Biology, Research Institute, National Center for Global Health and Medicine, Tokyo 162-8655, Japan. [11] Department of Chromosome Science, National Institute of Genetics, Research Organization of Information and Systems (ROIS), Yata 1111, Mishima, Shizuoka 411-8540, Japan. [12] Department of Genetics, The Graduate University for Advanced Studies (SOKENDAI), Yata 1111, Mishima, Shizuoka 411-8540, Japan. [13] Department of Molecular Microbiology, Research Institute for Microbial Diseases, Osaka University, Suita, Osaka 565-0871, Japan. [14] Advanced Research & Development Programs for Medical Innovation (PRIME), Japan Agency for Medical Research and Development (AMED), Chiyoda-ku, Tokyo 104-0004, Japan. [15] Cancer Cell Communication Project, NEXT-Ganken Program, Japanese Foundation for Cancer Research, Tokyo 135-8550, Japan. [16]These authors contributed equally: Sho Sugawara, Ryo Okada. ✉email: akiko.takahashi@jfcr.or.jp

Cellular senescence is an irreversible, arrested growth state induced by a variety of oncogenic stimuli. This important tumour suppressor mechanism prevents the proliferation of damaged cells at risk of becoming cancerous[1–4]. Senescent cells that accumulate with age gain the ability to produce various inflammatory proteins[5–7]. This phenomenon, which is caused by persistent DNA damage, termed as the senescence-associated secretory phenotype (SASP). SASP induces chronic inflammation in surrounding tissues and promotes numerous age-related pathologies such as cancer[8,9]. Senolytic drugs, which selectively target senescent cells, have been the subject of recent studies because the elimination of senescent cells from mouse models improved many age-related phenotypes and maintained homeostasis in tissues by inhibiting SASP[10–12]. Therefore, understanding the regulatory mechanisms underlying SASP induction has received attention with respect to the development of anti-ageing treatments.

In recent years, it has been demonstrated that activation of the innate immune response pathway through cyclic GMP–AMP synthase linked to a stimulator of interferon genes (cGAS-STING) signalling is important for SASP induction in senescent cells[13–17]. Endogenous genomic DNA ligands in the cytoplasm promote SASP through activation of the cGAS-STING pathway in senescent cells. However, the molecular mechanism of chromosomal DNA fragmentation and the origin of the nucleotide ligands for DNA sensors in senescent cells have not been explained.

Ribonuclease H2 subunit A (RNaseH2A) is the catalytic subunit of the RNase H2 endonuclease complex (RNaseH2A, RNaseH2B and RNaseH2C)[18]. Both RNase H1 and RNase H2 complexes resolve R-loop structures, consisting of double-stranded RNA-DNA hybrid and non-template single-stranded DNA, which are formed during transcription[19,20]. Unlike RNase H1, which can only process R-loops, RNase H2 can recognise ribonucleoside monophosphates (rNMPs) that have been incorporated into the genome because of DNA replication errors or DNA repair synthesis[21]. Accumulation of rNMPs has been considered a factor that induces genome instability[22–24]. RNASEH2A is responsible for Aicardi–Goutières syndrome (AGS)[25]. AGS is associated with a viral infection-like phenotype in the embryo and severe neurological symptoms in infancy, causing early childhood mortality[19], and the cGAS-STING pathway is activated in AGS model mice[26,27]. Accordingly, we hypothesised that RNaseH2A is involved in nucleotide ligand production and the subsequent induction of SASP during cellular senescence.

Several lines of evidence suggest that RNase H2 plays an important role in tumour progression. Intestinal epithelial-specific deletion of Rnaseh2b and Trp53 leads to the spontaneous development of small intestine and colon carcinomas in mice[28]. In addition, lower levels of RNase H2 are correlated with poor overall survival in patients with colorectal adenocarcinoma[28]. Another group demonstrated that RNASEH2B deletion is frequently detected in patients with metastatic prostate cancer or chronic lymphocytic leukaemia[29]. These reports indicate that RNase H2 functions as a tumour suppressor. However, the molecular interplay between a decrease in RNase H2 activity and gain of tumour malignancy has not been clarified.

In this study, we demonstrated that RNaseH2A gene expression is regulated by E2F transcription factors. Downregulation of RNaseH2A destabilises genomic DNA because of increased rNMP levels and facilitates nucleotide ligand accumulation, thereby activating cytoplasmic nucleic acid sensors in senescent and progeroid cells. This pathway induces inflammatory SASP-like gene expression and potentially contributes to the gain of malignant features in cancer cells.

## Results

### RNaseH2A expression and its enzymatic activity are suppressed in senescent cells

To identify potentially significant factors that regulate genomic DNA fragmentation and the production of nucleotide ligands for DNA sensor activation, we performed RNA sequencing (RNA-seq) using pre-senescent and senescent human diploid fibroblasts (HDFs; Supplementary Data 1 and 2). Pre-senescent TIG-3 and IMR-90 cells (early passage) were rendered senescent by either serial passage (late-passage; Fig. 1a and Supplementary Fig. 1a) or ectopic expression of oncogenic Ras (HRas$^{V12}$; Supplementary Fig. 1b). We investigated the changes in the expression of nuclease-related molecules involved in genomic DNA fragmentation in senescent HDFs. Based on gene set enrichment analysis (GSEA)[30,31], the gene expression associated with nuclease activity tended to be lower in senescent HDFs (Fig. 1a and Supplementary Fig. 1a, b). The top 10 downregulated genes in senescent cells compared to the findings in young cells are listed in Fig. 1b and Supplementary Data 3. Of these, Nei-like DNA glycosylase 3, a DNA glycosylase involved in telomere maintenance, was identified[32]. However, the knockout of Nei-like DNA glycosylases did not result in chromosomal instability or DNA fragmentation[33]. The next seven genes were primarily associated with homologous recombination repair, and it is known that homologous recombination decreases during cellular senescence[34]. Although the downregulation of RNA-degrading enzymes is related to genomic instability, its association with cellular senescence has not been fully elucidated. Therefore, we focused on RNaseH2A, an endonuclease that specifically degrades rNMPs in genomic DNA.

RNA sequence analysis revealed that the expression of RNase H family genes (RNaseH2A and RNaseH2B) was significantly decreased in senescent HDFs, and several senescence markers were detected (Fig. 1c). RNase H2 is a heterotrimer composed of the catalytic subunit RnaseH2A and non-catalytic subunits RNaseH2B and RNaseH2C. Of these genes, we confirmed a significant decrease in RNaseH2A expression via quantitative reverse transcription PCR (RT-qPCR; Fig. 1d and Supplementary Fig. 1c) and western blotting (Fig. 1e) in senescent HDFs. Coherently, senescent HDFs exhibited a significant reduction in RNase activity in an in vitro RNase H2-specific activity assay (Fig. 1f). These results indicate that RNaseH2A activity is significantly decreased in senescent cells.

### RNaseH2A expression is regulated by E2F transcription factors

To obtain insight into the mechanism responsible for the downregulation of RNaseH2A during cellular senescence, we conducted a promoter sequence analysis of human RNASEH2A. The transactivation activity of the E2F family is known to be repressed by the p16$^{INK4a}$-RB pathway in senescent cells[16,35,36]. Therefore, we hypothesised that E2F activity is crucial for RNASEH2A expression in HDFs. Based on this idea, we identified several putative binding sequences for the E2F-family transcription factors (Fig. 2a)[37]. Chromatin immunoprecipitation (ChIP) analysis illustrated that E2F transcription factors, namely E2F1 and E2F3, bound to the promoter region of RNASEH2A (Fig. 2b). To confirm the importance of E2F transcription factors in RNaseH2A expression, we used small interfering RNA (siRNA) to knock down DP1, an essential activator of E2F transcription factors[35,37]. This resulted in a significant reduction in RNaseH2A mRNA and protein expression in pre-senescent HDFs (Fig. 2c, d). Conversely, E2F3 overexpression increased the expression of RNaseH2A in HDFs (Supplementary Fig. 2a, b). Furthermore, a reporter analysis using deleted and mutated E2F-binding sites in the RNASEH2A promoter revealed the importance of E2F-binding sites for RNASEH2A expression (Fig. 2e, f and

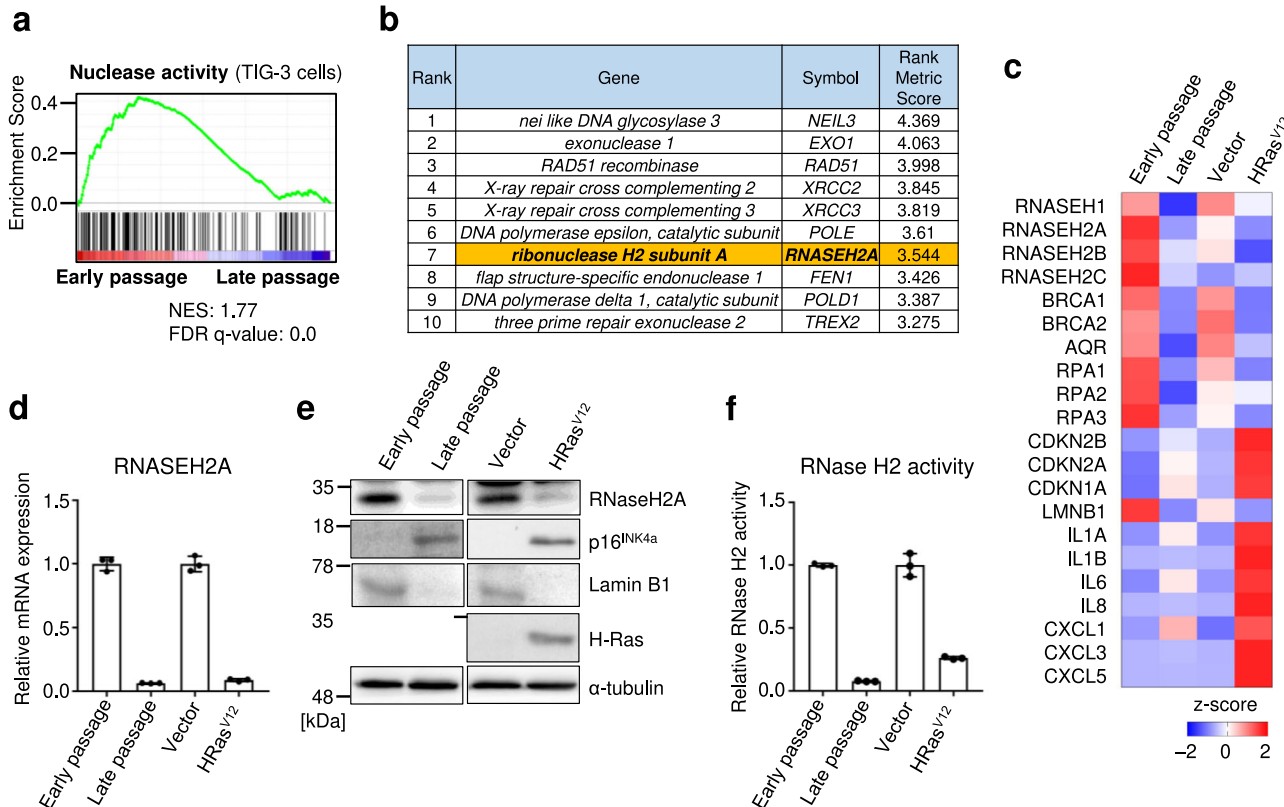

**Fig. 1 RNase H2 activity is decreased in senescent cells. a** Gene set enrichment analysis using the gene set for the nuclease activity of pre-senescent TIG-3 (early passage) and senescent TIG-3 cells (late-passage). NES: normalised enrichment score. FDR: false discovery rate. **b** List of the top ten genes from RNA sequencing. The rank reveals significantly downregulated genes in senescent TIG-3 cells compared to their expression in pre-senescent TIG-3 cells. See also Supplementary Data 1–3. **c** Heat map created by calculating the median from log2 (fragments per kilobase of transcript per million mapped reads) of early-passage, late-passage, or vector-infected and oncogenic Ras (HRas$^{V12}$)-infected TIG-3 cells. **d** Quantitative reverse transcription PCR analysis of RNASEH2A gene expression in early-passage, late-passage, vector-infected or HRas$^{V12}$-infected TIG-3 cells. Error bars indicate the mean ± SD of technical triplicates. **e** Western blot of TIG-3 cells as in **d** using the antibodies indicated on the right. Alpha-tubulin was used as a loading control. **f** In vitro RNase H2-specific activity assay of TIG-3 cells as in **d**. Error bars indicate the mean ± SD of technical triplicates. All data are representative of at least three biological replicates.

Supplementary Fig. 2c, d). According to ChIP-Atlas data[38], we confirmed the enrichment of E2F1 binding to the promoter region of *RNASEH2A* in three cancer cell lines (HeLa, MCF7 and MDA-MB231; Supplementary Fig. 2e). To further verify the significance of E2F and p16$^{INK4a}$-RB pathways, we treated HDFs with the cyclin-dependent kinase (CDK) 4/6 inhibitors palbociclib and abemaciclib, observing that CDK4/6 inhibition decreases *RNASEH2A* mRNA expression (Supplementary Fig. 2f). Collectively, these data indicate that RNaseH2A expression is regulated by E2F transcription factors.

**Residual rNMP levels in genomic DNA are increased in senescent cells.** Previous studies demonstrated that ribonucleotides misincorporated into the genome remained following *RNASEH2* knockout in yeast and mouse cells, leading to genomic DNA fragmentation[24,39–41]. Because RNase H2 activity was significantly reduced in senescent HDFs (Fig. 1f), we next examined whether rNMPs accumulate in the genomic DNA of senescent cells following a hydrolysis reaction under alkaline conditions using agarose gel electrophoresis. Compared to its mobility in control pre-senescent HDFs, genomic DNA from senescent HDFs exhibited significantly increased mobility following alkaline treatment (Fig. 3a, b), whereas its mobility was constant under normal conditions (without alkaline treatment; Supplementary Fig. 3a, b). We then estimated the distribution of the fragment sizes and the number of rNMPs incorporated (breakage) per cell

as previously described in refs. [24,40], finding that the length of the DNA fragments was significantly reduced and numerous ribonucleotides remained across the whole genome of senescent cells (Fig. 3c). To further investigate genomic DNA instability in senescent cells, we focused on cytoplasmic DNA fragments. A nucleic acid analyser detected an increase in short dsDNA fragments from a few to 100 bases in length in the cytoplasm of senescent cells (Supplementary Fig. 3c). In addition, we found that fragmented double-stranded DNA (dsDNA) signals significantly increased outside the nuclear membrane by super-resolution microscopic analysis (Supplementary Fig. 3d, e). Importantly, although RNaseH2A expression was reduced in quiescent cells, probably because of the decrease in E2F activity (Supplementary Fig. 3f, g), genomic DNA fragmentation was not observed by alkaline gel electrophoresis in quiescent cells (Supplementary Fig. 3h). These results suggest that genomic rNMPs and cytoplasmic dsDNA fragments are accumulated in senescent, but not quiescent, cells.

**RNaseH2A suppresses the expression of SASP factor genes.** To evaluate the importance of RNaseH2A for cellular senescence, we performed siRNA knockdown experiments. The depletion of RNaseH2A by siRNA in HDFs (siRNH2A#1 and #2 in TIG-3 and IMR-90 cells) resulted in increased SA-β-Gal activity (Fig. 4a, b and Supplementary Fig. 4a, b) and the accumulation of DNA damage, as previously reported[23,24,27,29,39] (Fig. 4c and

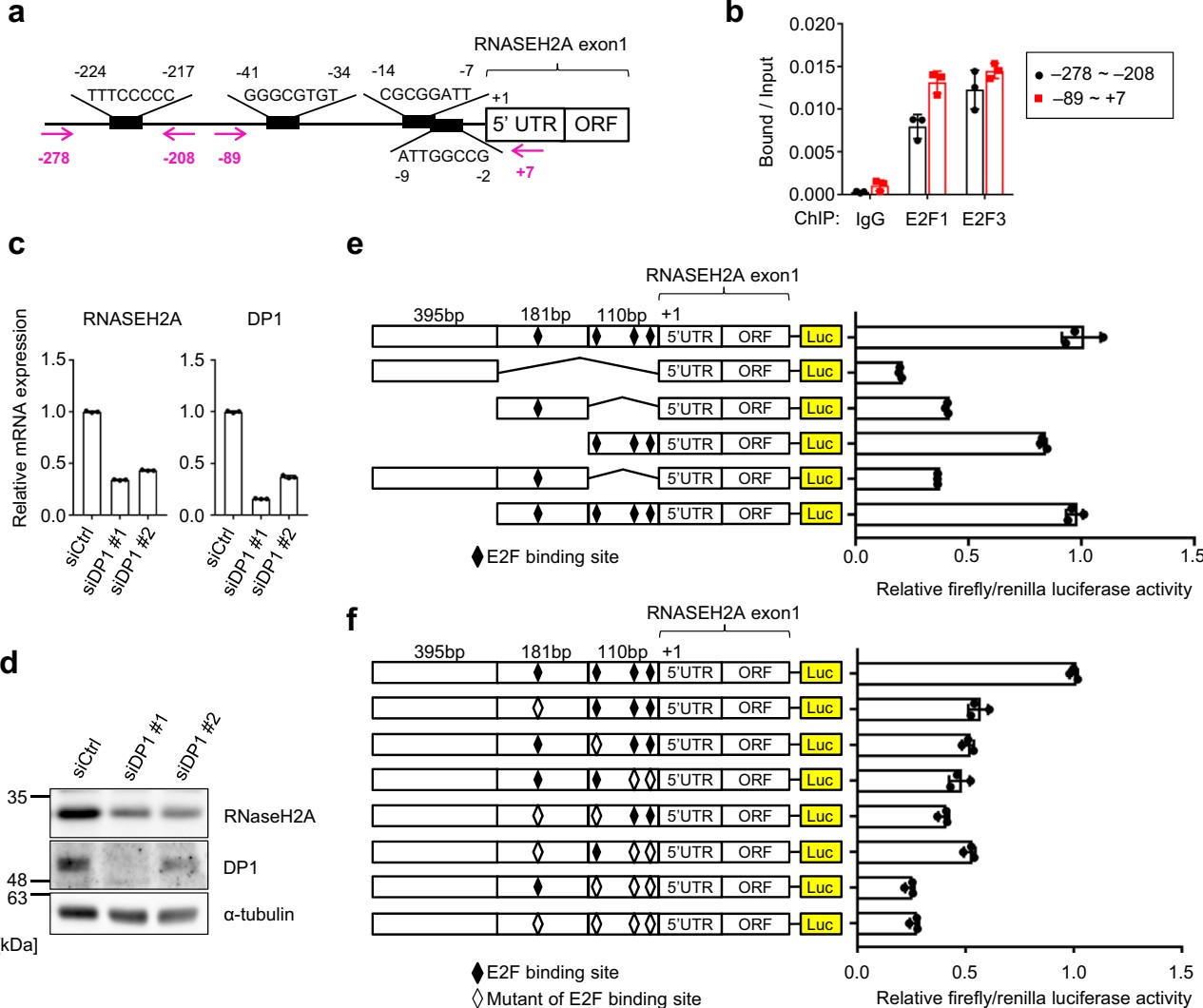

**Fig. 2 RNaseH2A is regulated by E2F transcription factors. a** Schematic diagram showing the E2F-binding sites in the RNaseH2A gene promoter. Red arrows indicate the region containing the PCR primers for ChIP analysis. **b** ChIP-qPCR analysis of pre-senescent TIG-3 cells using antibodies shown at the bottom. Error bars indicate the mean ± SD of technical triplicates. **c** RT-qPCR analysis of RNASEH2A and DP1 mRNA expression with siRNA-treated TIG-3 cells. Error bars indicate the mean ± SD of technical triplicates. **d** Western blot of TIG-3 cells as in **c** using antibodies shown at right. Alpha-tubulin was used as a loading control. **e** Left panel: Schematic representation of the reporter construct of human RNASEH2A gene promoter used in the analysis. Firefly luciferase is shown as Luc. Right panel: HEK293T cells were transfected with the indicated reporter plasmids and a Renilla plasmid as an internal control. After 48 h of transfection, luciferase activities were measured. Error bars indicate the mean ± SD of technical triplicates. **f** Left panel: Schematic representation of the reporter construct of human RNASEH2A gene promoter used in the analysis. The E2F-binding element or its mutant is presented as a black or white rhombus. The firefly luciferase is shown as Luc. Right panel: HEK293T cells were transfected with the indicated reporter plasmids and a Renilla plasmid as an internal control. After 48 h of transfection, luciferase activities were measured. Error bars indicate the mean ± SD of technical triplicates. All data were representative of at least three biological replicates.

Supplementary Fig. 4c). In addition, genomic DNA fragments were detected at high levels in the cytoplasm of RNaseH2A-depleted HDFs (Fig. 4d). Moreover, we discovered a significant decrease in the expression of LMNB1 and induction of the CDK inhibitor gene CDKN2A and various SASP factor genes, indicating that RNaseH2A depletion results in cellular senescence (Fig. 4e and Supplementary Fig. 4d). Additionally, cGAS or STING depletion significantly attenuated the induction of SASP factors caused by RNaseH2A depletion (Fig. 4f, g). These results indicate that SASP induction by RNaseH2A depletion largely depends on the cGAS-STING pathway.

Conversely, RNaseH2A overexpression restored RNase H2 activity and inhibited the accumulation of genomic DNA fragments in the cytoplasm in Ras$^{V12}$-induced senescent cells

(Fig. 5a–c and Supplementary Fig. 5a). Consistent with these data, SASP-related gene expression was attenuated by RNaseH2A overexpression in HRas$^{V12}$- and serial passage-induced senescent TIG-3 cells (Fig. 5d and Supplementary Fig. 5b). Collectively, RNaseH2A suppresses SASP factor gene expression.

**RNaseH2A is dysfunctional in cells derived from patients with Werner syndrome (WS).** To determine whether RNaseH2A is downregulated during ageing in vivo, we conducted a database analysis of RNA sequences derived from the skin fibroblasts of healthy donors of various ages[42] and found that RNASEH2A expression tends to decrease with age (Supplementary Fig. 6a). In addition, RNaseH2A was present at high levels in hepatic and

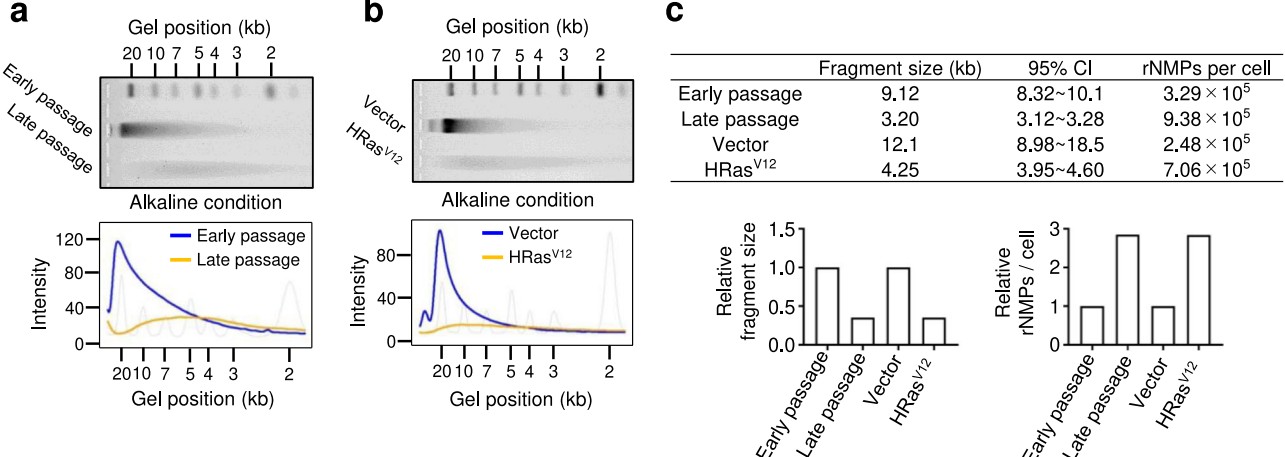

**Fig. 3 Residual ribonucleoside monophosphate levels in genomic DNA are increased in senescent cells. a** Top: Genomic DNA was extracted from pre-senescent (early-passage) or senescent (late-passage) TIG-3 cells and then subjected to alkaline gel electrophoresis following alkaline hydrolysis. The representative gel is shown from three biological replicates. Bottom: The density curves of the gel image are shown. **b** Top: Genomic DNA was extracted from pre-senescent (vector) or ectopic expression of oncogenic *ras* (HRas$^{V12}$) TIG-3 cells and then subjected to alkaline gel electrophoresis following alkaline hydrolysis. The representative gel is shown from three biological replicates. Bottom: The density curves of the gel image are shown. **c** The fragment sizes and numbers of ribonucleoside monophosphates (rNMPs) for each condition were estimated by the mathematical model with respect to representative data.

pancreatic cells collected from young mice; however, RNaseH2A expression was significantly decreased in p21$^{WAF1/CIP1}$-expressing cells, considered as senescent cells[12,16,43–45], in aged mouse tissues (Supplementary Fig. 6b–d). These results suggest that RNaseH2A expression is downregulated during the ageing process in vivo.

Mutations in *RNASEH2A* occur in AGS[19,25,41]; therefore, we speculated that the ablation of RNaseH2A expression may be associated with pathological phenotypes via SASP factor secretion in some age-related diseases. Because *RNASEH2A* expression decreases with age, we examined RNaseH2A expression in patients with progeria. WS is caused by a homozygous mutation in the RecQ-type DNA helicase (WRN helicase) on the short arm of chromosome 8, resulting in multiple aging phenotypes after the age of 40 years[46]. When we conducted GSEA using RNA sequence data from cells derived from patients with WS, the expression of a set of nuclease activity genes tended to be lower than that in normal fibroblasts (TIG-3 or NF; Supplementary Fig. 7a, b and Supplementary Data 3), as similarly observed in senescent cells[47,48] (Fig. 1a). Next, we examined the expression of RNaseH2A, observing clear downregulation in fibroblasts derived from patients with WS (WF1A, WF5 and WF8L) compared to its expression in cells from healthy volunteers of the same generation (NF1 and NF8-2) or established human fibroblast lines (TIG-3, IMR-90, BJ and Hs68; Fig. 6a, b and Supplementary Fig. 7c). Consistently, RNase H2-specific enzyme activity was also reduced in cells derived from patients with WS (Fig. 6c and Supplementary Fig. 7d). Agarose gel electrophoresis under alkaline conditions revealed that ribonucleotides were frequently incorporated into the genomes of cells derived from patients with WS and in senescent cells (Fig. 6d, e and Supplementary Fig. 7e). As expected, genomic DNA fragments accumulated in the cytoplasm of cells derived from these patients (Fig. 6f and Supplementary Fig. 7f). Moreover, SASP factor gene expression was upregulated in cells derived from patients with WS (Fig. 6g and Supplementary Fig. 7g). Furthermore, knockdown of DP1 in WF1A also resulted in decreased in RNaseH2A expression, indicating that RNaseH2A expression is regulated by E2F transcription factors in fibroblasts derived from patients with WS (Supplementary Fig. 7h, i). These results suggest that RNaseH2A downregulation is associated with age-related pathologies in patients with progeria.

**RNaseH2A depletion induces SASP-like gene expression in cancer cells.** Although a previous report suggested that intestinal epithelial *Rnaseh2b* deficiency in mice leads to severe DNA damage and that concomitant deletion of *Trp53* provokes spontaneous carcinogenesis[28], the underlying mechanism was not investigated in detail. We then hypothesised that RNaseH2A downregulation also induces inflammatory gene expression, which leads to the metastatic transformation of cancer. To test this hypothesis, we evaluated the effect of RNaseH2A ablation on SASP-like inflammatory gene expression using a degron-mediated protein degradation system in the HCT116 colorectal cancer cell line[49] (Fig. 7a). The degradation of RNaseH2A protein resulted in the accumulation of genomic DNA fragments in the cytoplasm (Fig. 7b, c). Coherently, the induction of some tumourigenic genes was observed in HCT116 cells (Fig. 7d). The SASP factor matrix metalloproteinase-2 can promote cell migration and invasion, contributing to cancer metastasis. Indeed, the downregulation of RNaseH2A significantly enhanced the invasiveness of HCT116 (Fig. 7e). Consistently, RNaseH2A knockdown also tended to increase SASP factor gene expression and invasion capacity in SK-OV-3 ovarian cancer cells (Supplementary Fig. 8a, b). These data suggest that RNaseH2A downregulation leads to malignant conversion in cancer cells. To further test this idea, we performed experiments using colonic organoids from colorectal cancer multistage carcinogenic model mice (Fig. 7f)[50]. Consistent with a gradual decline in RNaseH2A expression, cytoplasmic DNA accumulation and tumourigenic gene expression were observed in accordance with the degree of malignancy (Fig. 7g–i).

Finally, we found a strong correlation between RNASEH2A and E2F1 expression and an inverse correlation between RNaseH2A expression and poor prognosis in patients with colon, cervical, or ovarian cancer by an analysis of TCGA database[51] (Fig. 8a, b). Together, these data suggest that RNaseH2A downregulation results in the production of cytoplasmic nucleotide ligands, inducing

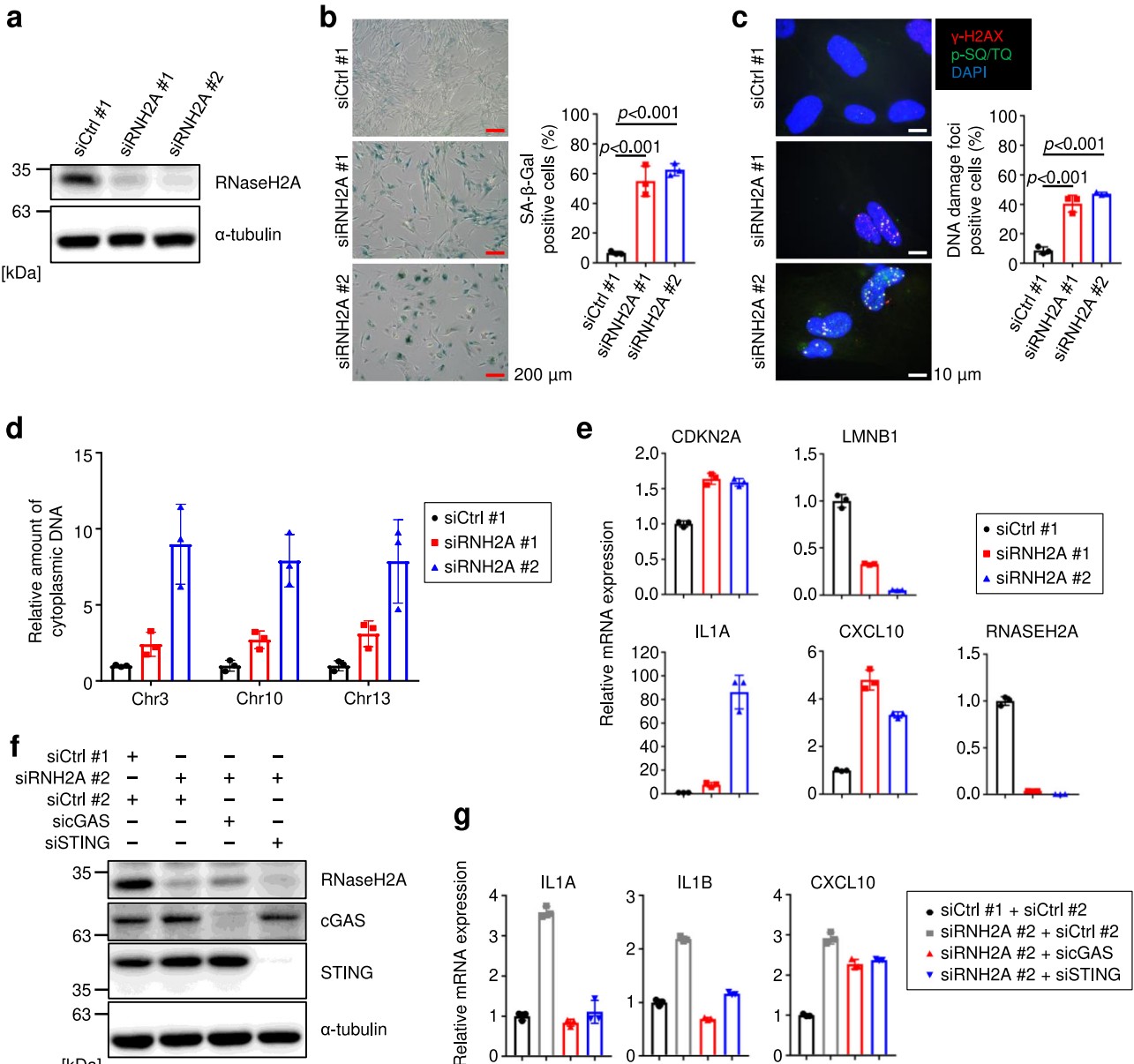

**Fig. 4 RNaseH2A depletion induces cellular senescence in human fibroblast. a** Western blot of pre-senescent TIG-3 cells treated with siRNAs (siRNH2A#1 and #2) for 72 h four times. Alpha-tubulin was used as a loading control. **b** Left: Representative images SA-β-Gal staining of TIG-3 cells as in **a**. Scale bars indicate 200 µm. Right: Quantification of SA-β-Gal positive cells. The graphs indicate the percentage of SA-β-Gal positive cells. Error bars indicate the mean ± SD of biological triplicates. One-way ANOVA coupled with Dunnett's multiple comparisons test. **c** Left: Representative immunofluorescence images of DNA damage marker of TIG-3 cells as in **a**. γ-H2AX (red), phospho-Ser/Thr ATM/ATR (p-SQ/TQ) substrate (green) and DAPI (blue). Scale bars indicate 10 µm. Right: Quantification of DNA damage-positive cells. The graphs indicate the percentage of nuclei containing more than two foci positive for both γ-H2AX and p-SQ/TQ. Error bars indicate the mean ± SD of biological triplicates. One-way ANOVA coupled with Dunnett's multiple comparisons test. **d** Quantitative PCR analysis of chromosomal DNA in the cytoplasm of TIG-3 cells as in **a** using primers against chromosomes 3, 10 and 13. Error bars indicate the mean ± SD of technical triplicates. **e** RT-qPCR analysis of the indicated genes using RNA extracted from TIG-3 cells as in **a**. Error bars indicate the mean ± SD of technical triplicates. **f** Western blot of TIG-3 cells depleted of RNaseH2A, cGAS or STING by siRNA. Alpha-tubulin was used as a loading control. **g** RT-qPCR analysis of indicated genes using RNA extracted from TIG-3 cells as in **f**. Error bars indicate the mean ± SD of technical triplicates. All data were representative of at least three biological replicates.

tumourigenic gene expression in cancer cells, which is likely linked to a poor prognosis.

## Discussion

Paradoxically, cellular senescence features both physiological and pathological aspects[2–4,10]. Although senescent cell cycle arrest can prevent tumour development, senescent cells also promote transformation, cancer invasion, and metastasis through the secretion of SASP factors[2–4,6]. SASP components have been reported to have both beneficial and harmful effects on human health, and these effects are regulated by multiple steps[2,4,6,52,53]. Remarkably, it was recently reported that SASP factor gene expression can be induced via the cGAS-STING pathway[13–17]. In this study, we identified the novel mechanism that RNaseH2A is transcriptionally regulated by E2Fs, thereby permitting rNMPs to

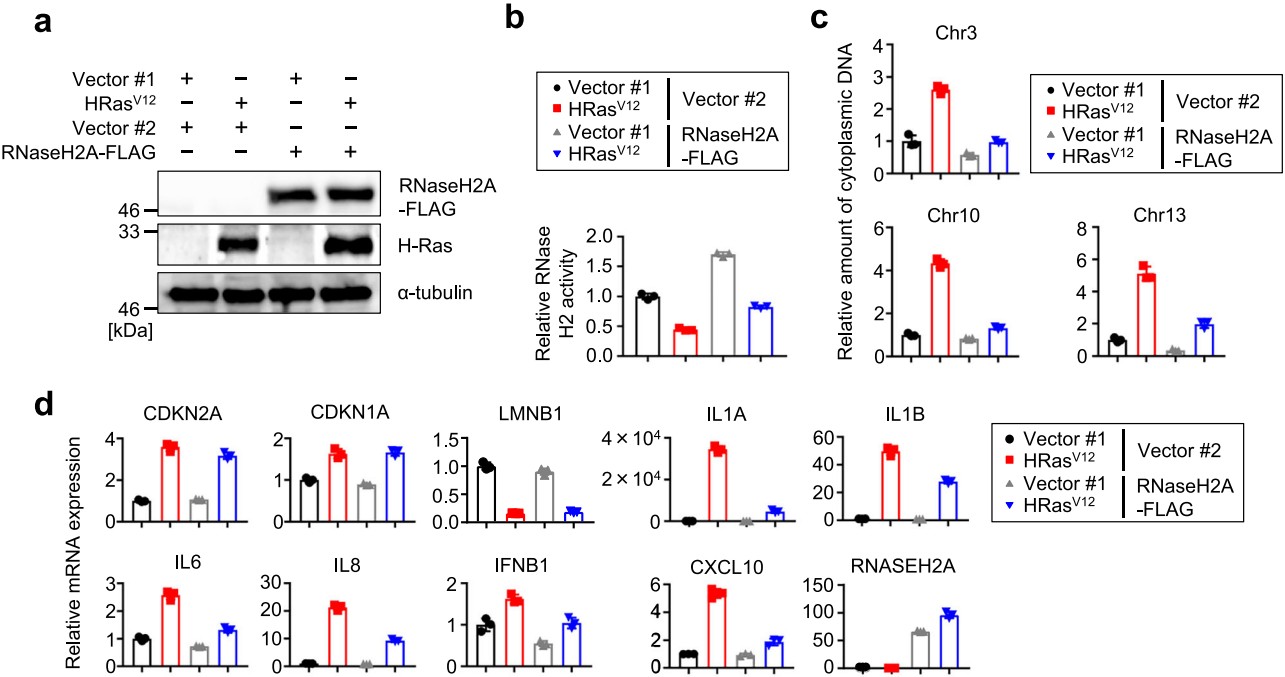

**Fig. 5 RNaseH2A suppresses the expression of SASP factor genes. a** Western blot of TIG-3 cells infected with retrovirus encoding flag-tagged RNaseH2A (lanes 3 and 4) or empty vector (lanes 1 and 2). After puromycin selection, cells were infected with retrovirus encoding HRas^V12 and subjected to western blot analysis using the antibodies shown at right. Alpha-tubulin was used as a loading control. **b** In vitro RNase H2-specific activity assay of TIG-3 cells as in **a**. Error bars indicate the mean ± SD of technical triplicates. **c** Quantitative PCR analysis of chromosomal DNA in the cytoplasm from TIG-3 cells as in **a** using primers against chromosomes 3, 10 and 13. Error bars indicate the mean ± SD of technical triplicates. **d** RT-qPCR analysis of the indicated genes was conducted using RNA extracted from TIG-3 cells as in **a**. Error bars indicate the mean ± SD of technical triplicates. All data were representative of at least three biological replicates.

promote genomic fragility via the downregulation of RNaseH2A during cellular senescence, leading to increased levels of genomic DNA fragments in the cytoplasm and the induction of SASP factors in senescent, progeroid and cancer cells (Fig. 8c).

The pathological features observed in individuals with WS, including ocular cataracts, dyslipidemia, diabetes mellitus, osteoporosis, atherosclerosis, premature hair greying with alopecia, and refractory skin ulcers might result from the pro-inflammatory function of SASP. In this study, we observed that RNaseH2A expression was also decreased in fibroblasts from patients with WS. Most importantly, patients with WS experience concomitant oncogenic complications[46]. It is possible that SASP-like gene expression contributes to the pathogenesis and onco-genesis of WS. We also observed that a decline in RNaseH2A expression is associated with the accumulation of genomic DNA fragments in the cytoplasm and oncogenic SASP-like gene expression in cancer cells and organoids. Furthermore, we found a strong correlation between E2F1 and RNaseH2A expression and an inverse correlation between RNaseH2A expression and poor prognosis in patients with colorectal, cervical, or ovarian cancer via TCGA database analysis[51]. In addition to a previous report regarding colorectal cancers[28], we also found the possibility that RNaseH2A ablation is associated with malignant phenotypes in other cancer types. Thus, further analysis is important to determine the relationship between the biological function of RNaseH2A and malignant alterations in various cancer types.

The accumulation of rNMPs in genomic DNA is associated with DNA damage and genomic instability; thus, *Rnaseh2*-knockout mice are non-viable from the embryonic stage[24,39]. Some progeria syndromes and age-related diseases may be suppressed by the regulation of nucleotide ligand production. In recent years, several cGAS and STING inhibitors have been

reported to suppress the innate immune response[54,55]. Consistent with these reports, we suggest that the nucleotide ligands for DNA sensors represent ideal targets for SASP regulation. Our study demonstrated that RNaseH2A is a key factor for the induction of SASP via nucleotide ligand accumulation during cellular senescence. Therefore, we anticipate that further analysis will reveal more details of this mechanism and lead to the discovery of novel strategies for SASP regulation.

## Methods

**Cell culture.** TIG-3, IMR-90, Hs68 and HEK293T cells were obtained from the JCRB. BJ and SK-OV-3 cells were obtained from ATCC. These cells were cultured in Dulbecco's Modified Eagle's medium (DMEM) supplemented with 10% foetal bovine serum (FBS). HCT116 cells expressing OsTIR1 and AID-RNaseH2A were cultured in McCoy's 5 A medium (Thermo Fisher Scientific, 16600082) supplemented with 10% FBS and 2 mM L-glutamine. To induce the degradation of AID-fused RNaseH2A protein, 500 µM indole-3-acetic acid (SIGMA, I2886), which is a natural auxin, and 2 µg ml⁻¹ doxycycline (Clontech, 631311) were added to the culture medium.

**Senescence or quiescence induction.** Early-passage TIG-3 or IMR-90 cells (fewer than 40 population doublings) were used as growing cells. Late-passage TIG-3 (more than 70 population doublings) or IMR-90 cells (more than 60 population doublings) that ceased proliferation were used as replicative senescent cells. For retroviral infection, TIG-3 cells were rendered sensitive to infection by ecotropic retroviruses. The cells were then infected with recombinant retroviruses encoding Ras^V12 (in pBabe-puro[56]) or RNaseH2A (in pMarX-puro[57]) cDNA. After the selection of infected cells with puromycin, pools of drug-resistant cells were analysed 7 days after infection. For X-ray–induced senescence, IMR-90 cells were exposed to 15 Gy irradiation using a CP-160 X-ray machine (Faxitron X-ray Corporation). After irradiation, IMR-90 cells were plated at a density of 2500 cells cm⁻² and analysed 10 days after irradiation. For quiescence induction, TIG-3 cells were cultured in DMEM supplemented with 0.1% FBS for 48 h. We confirmed the absence of mycoplasma contamination in the cultured cells.

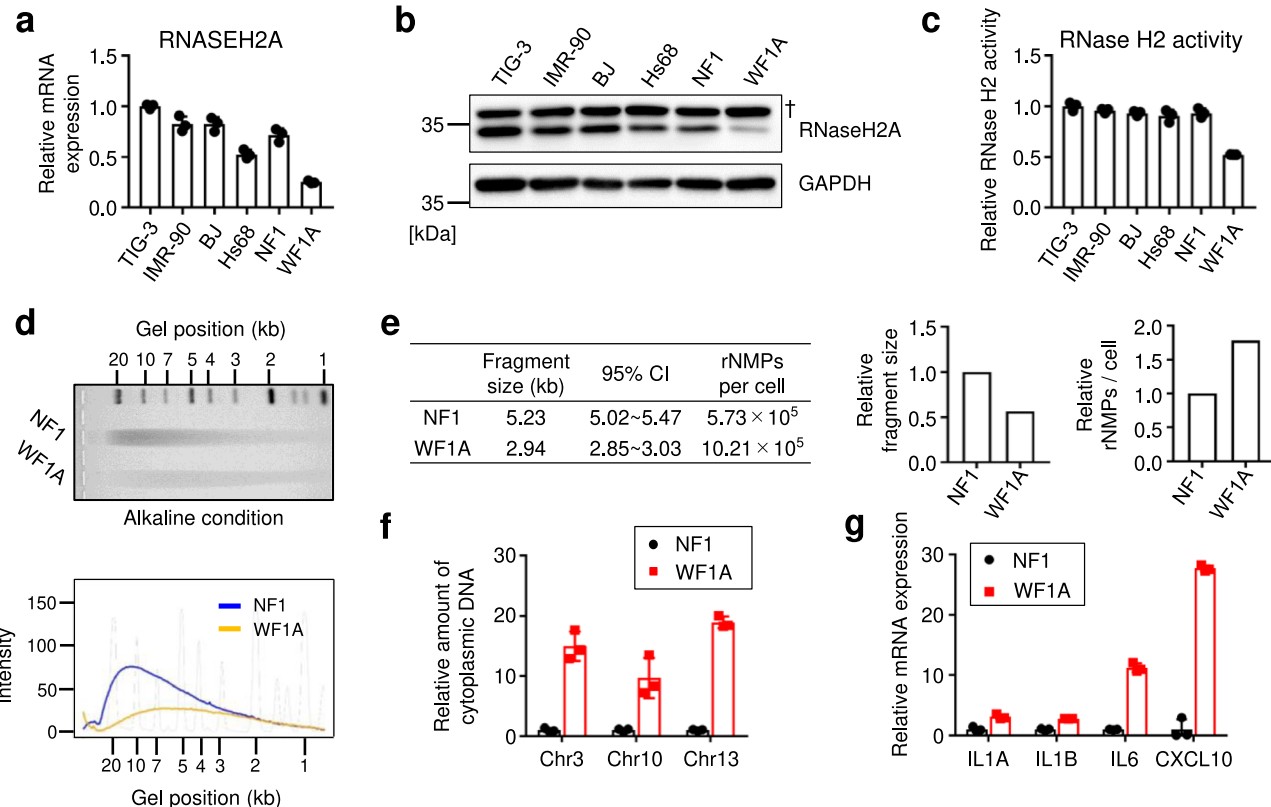

**Fig. 6 RNaseH2A dysfunction also occurs in cells derived from patients with Werner syndrome. a** RT-qPCR analysis of RNASEH2A mRNA expression in TIG-3, IMR-90, BJ, Hs68, NF1 and WF1A cells (35-40 population doublings). NF1: healthy human fibroblasts, WF1A: fibroblasts derived from patients with Werner Syndrome (WS). Error bars indicate the mean ± SD of technical triplicates. **b** Western blot of six human fibroblasts as in **a** using antibodies shown at right. Alpha-tubulin was used as a loading control. †: Nonspecific signal. **c** In vitro RNase H2-specific activity assay of six human fibroblasts as in **a**. Error bars indicate the mean ± SD of technical triplicates. **d** Top: Genomic DNA was extracted from NF1 or WF1A cells and then subjected to alkaline gel electrophoresis following alkaline hydrolysis. The representative gel is shown from three biological replicates. Bottom: The density curves of the gel image are shown. **e** The estimated fragment sizes and numbers of rNMPs in the NF1 and WF1A cells. **f** Quantitative PCR analysis of chromosomal DNA in the cytoplasm from the NF1 and WF1A cells using primers against chromosomes 3, 10 and 13. Error bars indicate the mean ± SD of technical triplicates. **g** RT-qPCR analysis of the indicated genes was conducted using RNA extracted from the NF1 and WF1A cells. Error bars indicate the mean ± SD of technical triplicates. All data are representative of at least three biological replicates.

**Plasmids**. The epitope-tagged cDNAs of RNaseH2A were cloned into the pMarX-puro retroviral vector. To amplify flag-tagged RNaseH2A cDNA, the following primers were used: 5'-ACCGGATCCACCGCCATGGATCTCAGCGAGCTGGA-3' (forward) and 5'-ACCGAATTCGAGGCTGGTTGCTGACTCC-3' (reverse). All cDNAs were sequenced using Genetic Analyzer 3130 (Applied Biosystems) and a BigDye Terminator v3.1 Cycle Sequencing Kit (Applied Biosystems).

**Isolation of cytoplasmic DNA fractions**. Cytoplasmic DNA was prepared by modifying a previously described method[16,58]. Briefly, cells were centrifuged for 1 min in a microcentrifuge, resuspended in 0.3 M sucrose buffer and homogenised by pipetting. The homogenate was overlaid with the same amount of 1.5 M sucrose buffer and centrifuged at $18,506 \times g$ for 10 min. Cytoplasmic DNA was purified by treatment with 0.4 mg ml$^{-1}$ proteinase K (Wako, 160-22752), phenol–chloroform extraction, and ethanol precipitation with a carrier (Dr. GenTLE® Precipitation Carrier, TaKaRa Bio Inc., 9094). The amount of nuclear DNA was measured via real-time PCR using three different sets of primers designed for different chromosomes (human chromosomes 3, 10 and 13). The fragment length of cytoplasmic DNA was defined using an NGS 3 K Reagent Kit and a LabChip GX Touch nucleic acid analyser (PerkinElmer).

**Immunofluorescence microscopy**. The cells were fixed with 4% paraformaldehyde/ PBS (Wako, 163-20145) and permeated with 0.5% Triton X-100/Tris-buffered saline for 1 min. The cells were blocked with 1% bovine serum albumin (BSA; Sigma-Aldrich, A3059) and 10% goat serum (Sigma-Aldrich, G9023)/ Tris-buffered saline for 1 h at 4 °C. The samples were then incubated with primary antibodies targeting γ-H2AX (1:1000, Millipore, 05-636) and phospho-(Ser/Thr) ATM/ATR substrate (1:500, Cell Signaling Technology, 2851). After incubation with secondary antibodies coupled to Alexa Fluor 488 or Alexa Fluor 594 (Thermo Fisher Scientific), the nucleus was stained with DAPI (Dojindo, 342-07431). After immunostaining, DNA damage-positive cells were quantified using a fluorescence microscope (Carl Zeiss).

**Super-resolution microscopic analysis**. The cells were fixed with 10% formalin (Wako) and 0.1% glutaraldehyde (SIGMA, G5882)/cytoskeleton stabilisation buffer (CSB) and permeated through the membrane with 0.5% Triton X-100/CSB for 1 min. CSB contained 137 mM NaCl, 5 mM KCl, 1.1 mM Na$_2$HPO$_4$, 0.4 mM KH$_2$PO$_4$, 4 mM NaHCO$_3$, 2 mM MgCl$_2$, 5.5 mM glucose, 2 mM EGTA and 5 mM PIPES (pH 6.1). The cells were blocked with 3% BSA/CSB for 1 h at 4 °C. NaBH$_4$ was used to reduce the fluorescent background caused by glutaraldehyde. Then, samples were reacted with primary antibodies targeting the dsDNA marker (1:500, HYB331-01, sc-58749) and lamin B1 (1:100, Abcam, ab16048). After incubation with antibodies conjugated to Alexa Fluor 488 or Alexa Fluor 555, the nucleus was stained using DAPI. After immunostaining, fluorescence images were observed and photographed using a TCS SP8 STED 3X super-resolution microscope (Leica Microsystems). dsDNA signals outside and associated with lamin B1 signals were regarded as fragmented dsDNA signals.

**RNase H2 activity assays**. RNase H2-specific activity assays were performed using a FRET-based fluorescent substrate release assay[39]. We annealed 10 μM 3'-FAM labelled oligonucleotides (GATCTGAGCCTGGGaGCT for RNase H2-specific activity; uppercase DNA, lowercase RNA) to a complementary 5'-DAB-CYL-labelled DNA oligonucleotide (NIPPON GENE CO., LTD.) in 60 mM KCl and 50 mM Tris-HCl (pH 8) by heating for 5 min at 95 °C followed by slow cooling to room temperature. Reactions were performed in 100 μl of buffer (60 mM KCl, 50 mM Tris-HCl [pH 8], 10 mM MgCl$_2$, 0.01% BSA, 0.01% Triton X-100) con-taining 0.25 μM oligonucleotide duplex in 96-well flat-bottomed plates at 37 °C for 3 h. In total 5 μg of each whole-cell lysate were used for each reaction. Fluorescence was read for 100 ms using a VICTOR Nivo Multimode Microplate Reader (PerkinElmer), with 480-nm excitation and 535-nm emission filters.

**Western blot analysis**. For western blotting, cells were lysed in lysis buffer (50 mM HEPES [pH 7.5], 150 mM NaCl, 1 mM EDTA, 2.5 mM EGTA, 10%

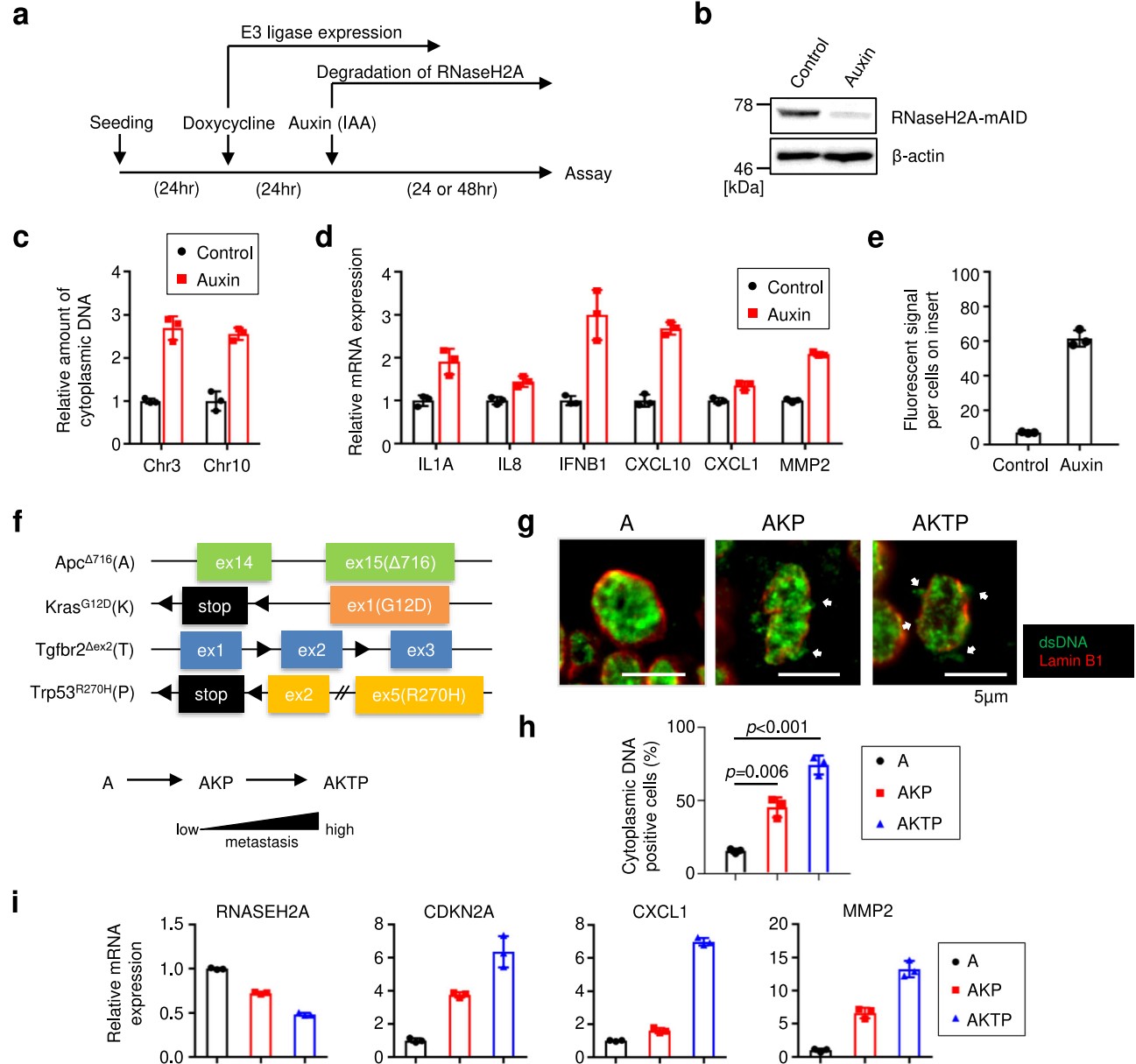

**Fig. 7 RNaseH2A depletion leads to malignant transformation in cancer cells. a** Scheme of experiments with HCT116 cells stably expressing TIR1 and RNaseH2A-mAID. Cells were treated with 500 µM auxin to degrade RNaseH2A-mAID for 24 h (**b**, **c**, **e**) or 48 h (**d**). **b** Western blot of HCT116 cells as in **a** using antibodies shown at right. Beta-actin was used as a loading control. **c** Quantitative PCR analysis of chromosomal DNA in the cytoplasm from HCT116 cells as in **a** using primers against chromosomes 3 and 10. Error bars indicate the mean ± SD of technical triplicates. **d** RT-qPCR analysis of the indicated genes was conducted using RNA extracted from HCT116 cells as in **a**. Error bars indicate the mean ± SD of technical triplicates. **e** Cell invasion assay using HCT116 cells as in **a**. Error bars indicate the mean ± SD of technical triplicates. **f** Top: Schematic representation of mutant alleles of each driver gene. Bottom: Schematic representation model of the evolutionary changes in the gene expression pattern in association with malignant progression phenotypes. **g** Colon cancer organoids were subjected to immunofluorescence for indicated antibodies. Scale bars indicate 5 µm. White arrows show dsDNA fragments in the cytoplasm. **h** The percentage of cytoplasmic DNA-positive cells in **g**. Error bars indicate the mean ± SD of biological triplicates. One-way ANOVA coupled with Dunnett's multiple comparisons test. **i** RT-qPCR analysis of the indicated genes was performed using RNA extracted from colon cancer organoids as in **g**. Error bars indicate the mean ± SD of technical triplicates. All data were representative of at least three biological replicates.

glycerol, 0.1% Tween 20 and 10 mM β-glycerophosphate) containing 1% protease inhibitor cocktail (Nacalai Tesque, 25955-11). The protein concentration was determined using the DC Protein Assay (Bio-Rad), and the proteins were separated via SDS-PAGE and then transferred to PVDF membranes (EMD Millipore). After blocking with 5% milk, membranes were probed with primary antibodies targeting HRas (Santa Cruz, sc-29), p16 (IBL, 11104), STING (Cell Signaling Technology, 13647), cGAS (Cell Signaling Technology, 15102), Lamin B1 (Abcam, ab16048), DP1 (Abcam, ab11834), α-tubulin (Sigma-Aldrich, T9026), RNaseH2A (PRO-TEINTECH, 16132-1-AP), E2F3 (Santa Cruz, sc-878), mini-AID-tag (MBL, M214-3), β-actin (Santa Cruz, sc-47778) and GAPDH (PROTEINTECH, 60004-1-Ig). The membranes were incubated with a mouse (GE Healthcare, NA931-1ML) or

rabbit secondary antibodies (GE Healthcare, NA934-1ML), visualised using SuperSignal West Femto Maximum Sensitivity Substrate (Thermo Fisher Scientific, 34096) and detected using FUSION SOLO S (Vilber Lourmat).

**Alkaline agarose gel electrophoresis.** Genomic DNA was purified from cultured cells by treatment with 0.4 mg ml$^{-1}$ proteinase K, phenol–chloroform extraction, and ethanol precipitation. For alkaline hydrolysis, 1 µg of genomic DNA was incubated with 0.3 M NaOH at 55 °C for 2 h. After alkaline treatment, 6× loading buffer (300 mM NaOH, 6 mM EDTA, 18% [w/v] Ficoll PM400, 0.15% bromocresol green and 0.25% xylene cyanol FF) was added to the treated DNA samples. Electrophoresis of alkaline-treated samples was performed using a gel containing

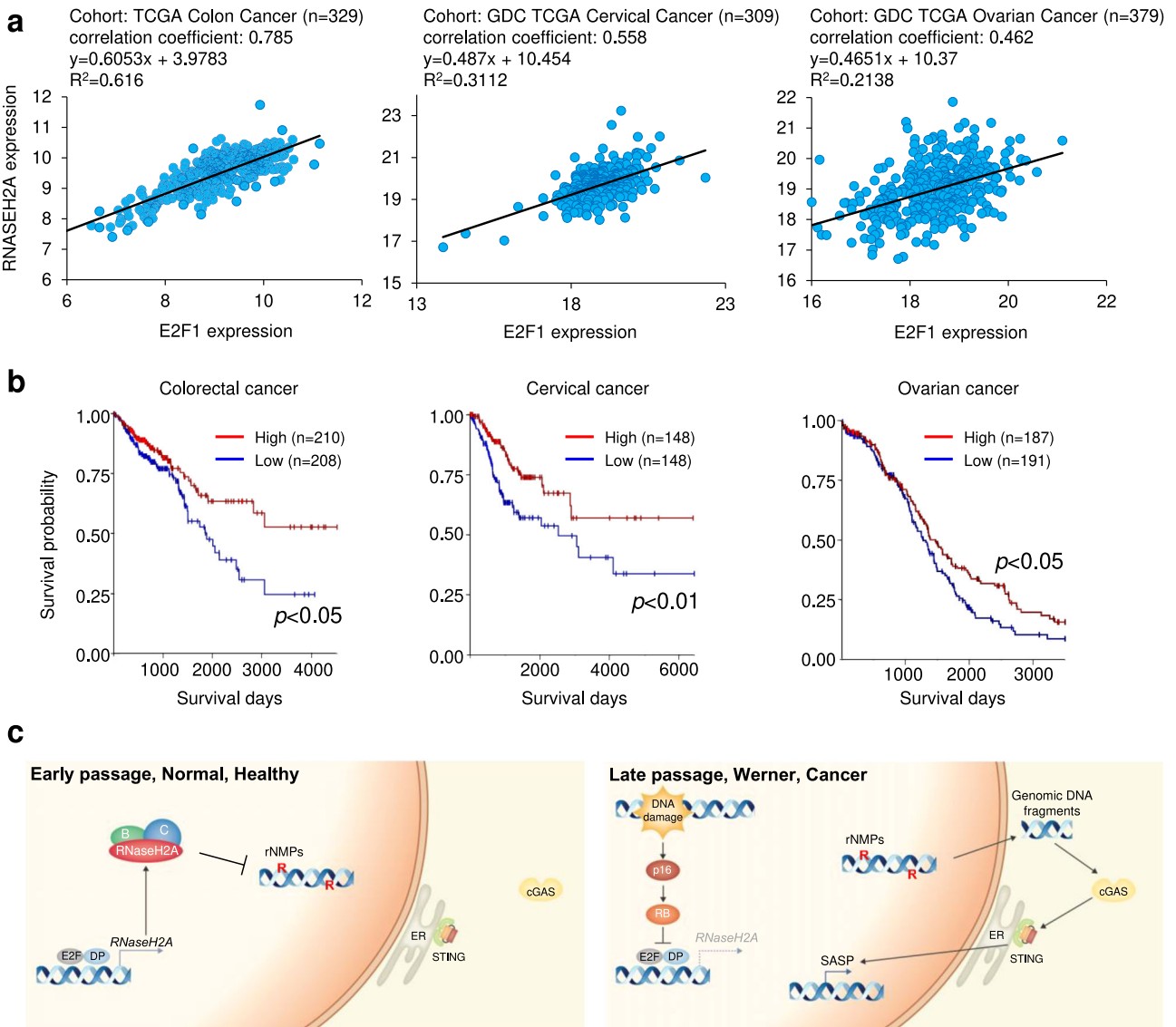

**Fig. 8 Low RNaseH2A expression is associated with poor prognoses in cancer. a** Plots of RNASEH2A and E2F1 mRNA expression levels in tissues derived from patients with colon ($n = 329$), cervical ($n = 309$) and ovarian ($n = 379$) cancers from TCGA database. The correlation between E2F1 expression and RNASEH2A expression was evaluated by Pearson's correlation coefficient. **b** Comparison of overall survival time between patients with low expression of RNaseH2A (blue) and those with high expression (red) in colorectal ($n = 418$), cervical ($n = 296$) and ovarian ($n = 378$) cancers from TCGA database. Log-rank test. **c** Outline the diagram of this paper.

50 mM NaOH, 1 mM EDTA and 0.8% w/v alkaline agarose and loading buffer containing 50 mM NaOH and 1 mM EDTA running buffer at 1 V cm$^{-1}$ for 18 h. After electrophoresis, the gel was neutralised in 1 M Tris-HCl buffer (pH 7.6) containing 1.5 M NaCl, stained with FluoroStain™ DNA Fluorescent Staining Dye (SMOBIO TECHNOLOGY, INC. DS1000), and visualised using FUSION SOLO S.

**Prediction of ribonucleotide incorporation rates**. Ribonucleotide incorporation rates were estimated using the same similar method as that described by ref. [24]. First, the densitometric images of the electrophoresis lanes corresponding to alkali-treated genomic DNA were taken. These images were pre-processed with GelJ 2.0 (Jónathan Heras, César Domínguez, Eloy Mata, César Larrea and Vico Pascual). The relevant lanes were cropped, the background was removed, and the densitometric curves were obtained. The densitometric curves were then analysed using R (R Development Core Team [2008]. R: a language and environment for statistical computing. R Foundation for Statistical Computing, Vienna, Austria. ISBN 3-900051-07-0, URL http://www.R-project.org.). The curves were smoothed by convoluting with a Gaussian filter (window size 3%) of the lane length. The conversion from the electrophoretic distance (d) to the fragment size (sz) was performed using the equation d = a + b log(sz), where coefficients a and b were estimated by fitting to the size reference (marker) lane. The histogram of the densitometric intensity was obtained at intervals of width $\Delta$sz = 0.5 mm. The number of fragments at each interval was assumed to be proportional to the

densitometric intensity divided by sz. Moreover, we assumed that the probability of break at a point was uniform along the genome so that the distribution of the fragment length could be approximated by the exponential distribution. We estimated the mean (m) of the distribution and by the property of exponential distribution, the number of incorporations (breakage) per genome was obtained as m_0/m, where m_0 denotes the original size of the genome.

**ChIP-qPCR analysis**. ChIP analysis was conducted using Dynabeads® Protein G (Thermo Fisher Scientific, 10004D) according to the manufacturer's protocols. Briefly, chromatin was extracted from TIG-3 cells, cross-linked with formaldehyde (1% final concentration), and sonicated (Bioruptor, Cosmo Bio Co. Ltd.: 10 cycles of 30 s on/30 s off, on the highest setting) to generate DNA fragments[52]. The immunoprecipitation of cross-linked chromatin was conducted using anti-human E2F1 (Santa Cruz, sc-193X), anti-human E2F3 (Santa Cruz, sc-878X), or rabbit IgG (Cell Signalling Technology, 2729) as a negative control. After immunoprecipitation, the DNA was extracted using a QIAquick PCR purification kit (Qiagen, 28104), and an aliquot was amplified via real-time PCR using primers flanking the putative human E2F-binding site position at −278 to −208 bp of the human *RNaseH2A* promoter (5′-CTAGTTCCCCGTTAGGCTGG-3′ and 5′-CGGTGGGAAGAGGGGGAAAT-3′) or primers flanking the putative human E2F-binding site position at −89 to +7 of the human *RNaseH2A* promoter (5′-GGCGTCGGCTGCGTAA-3′ and 5′-GCCTGCTTCCGGCCAAT-3′).

**RNA-seq analysis**. Total RNA from cultured cells was isolated using a mirVana Kit (Thermo Fisher Scientific, AM1560). DNA contamination was eliminated with DNase (TURBO DNA-free™ Kit, Invitrogen, AM1907) treatment. Library preparation was performed using a TruSeq mRNA Sample Prep. Kit (Illumina) (early passage, late-passage, TIG-3) or TruSeq Standard mRNA LT Sample Prep Kit (Illumina) (Vector, HRasV12, TIG-3). Sequencing was performed using the Illumina HiSeq2500 platform (early passage, late-passage, TIG-3) or the Illumina NovaSeq6000 platform (Vector, HRasV12, TIG-3). The quality of raw reads was evaluated using FASTQC v0.11.5 (http://www.bioinformatics.babraham.ac.uk/projects/fastqc). To reduce bias in the analysis, artifacts such as low-quality reads, adaptor sequence, contaminating DNA and PCR duplicates were removed using Trimmomatic 0.32 (http://www.usadellab.org/cms/?page=trimmomatic). Trimmed reads were mapped to the reference genome hg19 using TopHat v2.1.1 (https://ccb.jhu.edu/software/tophat/index.shtml) splice-aware aligner. The transcript was assembled using Cufflinks v2.2.1 (http://cole-trapnell-lab.github.io/cufflinks/) with aligned reads that contain paired-end information. The expression profiles of assembled transcripts for each sample were calculated using Cufflinks and represented as normalised values, which were based on transcript length and coverage depth. The fragments per kilobase of transcript per million mapped reads served as the expression profile.

**GSEA**. GSEA was conducted using the GSEA v4.1.0 programme (Broad Institute) with the RNA-seq or microarray data[30] gene set for nuclease activity. We used our RNA-seq data, proliferating, senescent IMR-90 cell microarray data (GSE36640)[31] and WS patient-derived fibroblast microarray data (GSE62114, GSE48761)[47,48].

**Luciferase-reporter assays**. The human RNaseH2A gene promoter sequence (NCBI Reference Sequence: NG_012662.1) was amplified with PCR using genomic DNA extracted from HDFs. The sequences of the PCR primer sets used to clone the RNaseH2A promoter were as follows: RNaseH2A-promoter-F, 5′-accggtaccTGGGGA-CATTCATTCTAAGTTGGGAGCCGCTCGAGATTACGATTCTTATGTGTGTG-3′ (containing a KpnI site [underlined]); and RNaseH2A-promoter-R, 5′-accacgcgtGTTTCCCGCATCCTCCGTAC-3′ (containing a MluI site [underlined]). Deletion mutants were prepared using standard PCR procedures. The sequences of the primer sets used to prepare deletion mutants of the RNaseH2A promoter were as follows: RNaseH2A-promoter-Cut1-F, (395 bp), 5′-accggtaccTGGGGA-CATTCATTCTAAGTTGGGAGCCGCTCGAGATTACGATTCTTATGTGTGTG-3′ (containing a KpnI site [underlined]); RNaseH2A-promoter-Cut1-R, 5′-accacgcgtGCTTCGAAGACCCAGCCTAACG-3′ (containing a MluI site [underlined]; RNaseH2A-promoter-Cut2-F, (181 bp) 5′-accggtaccCGTTAGGCTGGGTCTTC-GAAGC-3′ (containing a KpnI site [underlined]); RNaseH2A-promoter-Cut2-R, 5′-accacgcgtCTCCTGGGAATTGTAGTCCCGA-3′ (containing a MluI site [underlined]); RNaseH2A-promoter-Cut3-F (110 + 94, 181 bp), 5′-accggtaccTCGGGACTA-CAATTCCCAGGAG-3′ (introducing a KpnI site [underlined]); and RNaseH2A-promoter-Cut3-R, 5′- accacgcgtGTTTCCCGCATCCTCCGTAC-3′ (containing a MluI site [underlined]). Promoter sequences containing point mutations were generated using a QuickChange site-directed mutagenesis kit (Agilent Technologies). The sequences of the PCR primer sets used to generate point mutations for the RNaseH2A promoter were as follows: RNaseH2A-promoter-muation1-F, 5′-cagtttccctcttattttcAA-cAtcttcccaccgggccacg-3′ (containing a mutation site [underlined]); RNaseH2A-promoter-muation1-F, 5′-cgtggcccggtgggaagaTgTTgaaataagagggaaactg-3′ (containing a mutation site [underlined]); RNaseH2A-promoter-muation2,3-F, 5′-gtgttctgccAgAA-gattgAcAAgaagcaggcgccg-3′ (containing a mutation site [underlined]); and RNaseH2A-promoter-muation2,3-R, 5′-cggccgcctgcttcTTgTcaatcTTcTggcagaacac-3′ (containing a mutation site [underlined]). The promoter fragments were inserted into the pGL3 basic firefly luciferase-reporter plasmid (Promega). All inserted DNAs were sequenced and verified. Reporter plasmids were transfected into HEK293T cells using X-treamGENE9 DNA transfection reagent (Roche, 6365809001) according to the manufacturer's protocols. The luciferase assays were performed using a Luciferase Assay System Kit (Promega, E1910). Cytomegalovirus promoter Renilla luciferase plasmid was used as an internal control.

**RNA interference**. RNA interference was performed by transfecting siRNA oligos using Lipofectamine™ RNAiMAX transfection reagent (Thermo Fisher Scientific, 13778075), according to the manufacturer's protocols. The sequences of the siRNA oligos used in Fig. 2 and Supplementary Fig. S8 were as follows: siCtrl[59], GCGCGCUUUGUAGGAUUCG; siDP1 #1, ACGCCUCAGAGACCGGCAG; and siDP1 #2, AUGACCAGAAAAACAUAAG. The following ON-TARGETplus siRNAs (Dharmacon) were used in Fig. 4 and Supplementary Figs. 4, 8: siCtrl #1, ON-TARGETplus Non-targeting siRNA #3 (D001810-03); siRNH2A #1, ON-TARGETplus Human RNASEH2A siRNA (J-003535-09); siRNH2A #2, ON-TARGETplus Human RNASEH2A siRNA (J-003535-12); siCtrl #2, ON-TARGETplus Non-targeting Pool (D001810-10); sicGAS, ON-TARGETplus Human MB21D1 siRNA SMARTPool (L-015607-02); and siSTING, ON-TARGETplus Human TMEM173 siRNA SMARTPool (L-024333-00).

**RT-qPCR**. Total RNA was extracted from cultured cells using a mirVana miRNA Isolation Kit (Thermo Fisher Scientific) and then subjected to reverse transcription using a PrimeScript RT reagent kit (TaKaRa Bio Inc., RR037A). RT-qPCR was performed on a StepOnePlus PCR system (Applied Biosystems) using SYBR Premix Ex Taq (TaKaRa Bio Inc., RR820A). The PCR primer sequences are listed in Supplementary Data 4.

**Cell invasion assay**. The cell invasion assay was performed using CytoSelect™ 24-Well Cell Invasion Assay, Basement Membrane (Cell Biolabs, CBA-111) according to the manufacturer's instructions. Briefly, HCT116 cells expressing OsTIR1 and AID-RNaseH2A were incubated with 500 μM auxin (indole-3-acetic acid) after 2 μg ml⁻¹ doxycycline treatment for 24 h. After auxin treatment for 24 h, the cells were seeded at a density of $5 \times 10^5$ cells per polycarbonate membrane insert utilising 300 μl of medium containing 0.1% FBS. SK-OV-3 cells were treated with 5 nM siRNA three times for 72 h each and seeded on polycarbonate membrane inserts. Then, 500 μl of medium containing 10% FBS were added to the lower well of the plate, which was cultured for 24 h. Invasive cells passed through the basement membrane layer and clung to the bottom of the insert membrane. The cells were dissociated using Cell Detachment Buffer, lysed, and quantified using CyQuant GR fluorescent dye. Fluorescence was read for 100 ms using a VICTOR Nivo Multimode Microplate Reader with 480 nm excitation and 535 nm emission filters.

**SA-β-gal assay**. Cells were fixed in fixation buffer (2% PFA and 0.2% glutaraldehyde in PBS) and stained with staining solution (5 mM potassium ferricyanide, 5 mM potassium ferrocyanide, 2 mM $MgCl_2$, 150 mM NaCl and 1 mg/ml X-Gal) in citrate/sodium phosphate buffer (pH 6) overnight at 37 °C. Cells were then washed twice with PBS, and the percentage of stained cells was determined.

**Organoid experiments**. $Apc^{\Delta716}$, $Kras^{LSL \cdot G12D}$, $Tgfbr2^{flox/flox}$, $Trp53^{LSL/R270H}$ and villin-CreER mice have been described previously[50]. The organoid cultures were established from small intestinal tumours, as described previously[60]. We were provided with the small intestinal tumour organoids from $Apc^{\Delta716}$, $Kras^{LSL \cdot G12D}$, $Tgfbr2^{flox/flox}$, $Trp53^{LSL/R270H}$ and villin-CreER mice by Dr. Nakayama and Dr. Ohsima of Kanazawa University.

For passage, the organoids were removed from Matrigel (BD Biosciences) using Cell Recovery Solution (Corning, #354253) and mechanically separated into a single coding domain. The crypts were mixed with 30 μl of Matrigel and seeded in 48-well plates. After Matrigel polymerisation, growth factors (50 ng ml⁻¹ EGF [Thermo Fisher Scientific, PMG8041]), GlutaMax supplement (Thermo Fisher Scientific, 35050061), HEPES (Thermo Fisher Scientific, 15630106), N-2 supplement (Thermo Fisher Scientific, 17502048), B-27 supplement (Thermo Fisher Scientific, 17504044), N-acetyl-L-cysteine (SIGMA, A9165), and medium (Advanced DMEM/F12 (Thermo Fisher Scientific, 12634010)) were added. The entire medium was changed every week. Passages were performed every 1–2 weeks at a 1:5 split ratio.

Immunofluorescence analysis were performed as previously described in ref. [61] with minor modifications. Briefly, the organoids were fixed in 4% paraformaldehyde (Nacalai Tesque, 26126)/PBS at room temperature for 30 min, mechanically dissociated from Matrigel by pipetting, washed with PBS supplemented with 0.2% BSA and suspended in 2.0% agarose (SIGMA, A2676). Then the samples were centrifuged at 1000 rpm for 3 min at room temperature in 50-ml tubes, and the samples were cooled on ice, removed from the tubes, and embedded in paraffin. Samples were sectioned on a microtome (3-μm-thick), deparaffinized in xylene, rehydrated, and then analyzed via histological examination. For antibody staining, deparaffinized and rehydrated sections were subjected to heat-induced antigen retrieval for 20 min at 121 °C in sodium citrate buffer (10 mM sodium citrate, 0.05% Tween 20, pH 6.0) in an autoclave. After washing in PBS, the sections were incubated in a blocking buffer (10% goat serum/PBS) for 1 h at room temperature. Then, the sections were incubated with primary antibodies against lamin B1 and dsDNA marker in 1% BSA/PBS overnight at 4 °C. To detect primary antibodies, relevant Alexa Fluor 488 goat anti-mouse or Alexa Fluor 594 goat anti-rabbit secondary antibodies (Invitrogen, 1:500) were used. Fluorescence images were observed and photographed using a BZ-X700 immunofluorescence microscope (Keyence).

**Animal experiments**. Liver and pancreas tissues from young (6 weeks old) and aged (108 weeks old) mice were prepared using C57/BL6J mice (CLEA Japan, Inc). The biopsy samples were subjected to fluorescent immunohistochemistry using antibodies against p21$^{WAF1/CIP1}$ (BD Biosciences, 556430) and RNaseH2A (OriGene, TA306706), as previously described[62]. All animal procedures were performed using protocols approved by the Japanese Foundation for Cancer Research (JFCR) Animal Care and Use Committee in accordance with the relevant guidelines and regulations (approval number: 1804-05). Signal quantification of images was performed using the ImageJ software (NIH, v1.53).

**Clinical samples**. The use of clinical samples was approved by the Institutional Review Board of Chiba University Graduate School of Medicine (approval number: 1145). The samples were obtained during surgical procedures. The patients provided written informed consent before the surgery. NF1, NF8-2, WF1A, WF5 and WF8L cells are human skin fibroblasts. NF1 cells were collected from a healthy 42-year-old Japanese man and NF8-2 cells were collected from a healthy 45-year-old

Japanese man. WF1A cells were collected from a 47-year-old Japanese man diagnosed with WS, WF5 cells were collected from a 43-year-old Japanese man diagnosed with WS and WF8L cells were collected from a 43-year-old Japanese man diagnosed with WS. WF1A, WF5 and WF8L cells carried a homozygous mutation in the WRN gene (Mut4 mutation: c.3139-1 G > C). These cells were collected from patients who provided informed consent for genetic and cell biological analyses. The cells were cultured in DMEM supplemented with 10% FBS. All methods were performed in accordance with the protocols approved by the Institutional Review Board (approval number: 2019-1211) of JFCR.

**Statistics and reproducibility**. Statistical analysis was conducted using an unpaired two-tailed Student's *t*-test, one-way ANOVA coupled with Dunnett's multiple comparisons test, two-way ANOVA coupled with Tukey's multiple comparison test, or the log-rank test. Statistical analyses were carried out by PRISM software version 7.04. *p* values less than 0.05 were considered statistically significant. Error bars indicate means ± SD. Results were repeated at least three times unless indicated otherwise.

## Data availability

Nucleotide sequence data reported are available in the DDBJ Sequenced Read Archive under the accession numbers DRA009786. Source data are available in Supplementary Data 5. The data that support the findings of this study are available in the supplementary material of this article.

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

## Acknowledgements

We thank the members of the Takahashi laboratory for their helpful support during the preparation of this manuscript. This work was supported in part by grants from the Japan Agency of Medical Research and Development (AMED) under grant number 19gm6110023h0001; the Japan Science and Technology Agency (JST)-PRESTO under grant number JPMJPR17H7; JST-Moonshot R&D under grant number JPMJPS2022; Japan Society for the Promotion of Science (JSPS) under grant number (No. 17K19618, 19H03507, 17H06413, 17H06417, 18J10977, 21J01769, 22H02907 and 22K07198); the Naito Foundation, the Uehara Memorial Foundation and the Foundation for Promotion of Cancer Science.

## Author contributions

S.S., R.O. and A.T. designed the experiments and wrote the manuscript. S.S., R.O., T.M.L., H.T., K.M., M.C., H.K., K.K., S.K., Y.M., M.N., K.N., S.N., and K.T. performed the experiments. K.K., S.K., K.Y., M.O., C.O., A.N., M.T.K., E.H. and A.T. analysed the data. A.T. oversaw the projects.

## Competing interests

The authors declare no competing interests.
