## [Peer Review File · Communications Biology]

Reviewers' comments:

Reviewer #1 (Remarks to the Author):

This manuscript by Okada et al. describes a new mechanism whereby the senescence-associated secretory phenotype (SASP) is regulated. This is a highly important question as the SASP has major detrimental effects in both cancer and age-related pathologies. They find that RNaseH2A is decreased during cellular senescence and physiological and pathological aging. This was associated with increased genomic rNMP accumulation and cytoplasmic dsDNA, a known activator of the SASP. Using experiments to knockdown or degrade RNaseH2A, they also observe cytoplasmic dsDNA and SASP expression. This is an intriguing study that provides new insight into the upstream mediators of SASP activation.

Major Concerns

1. Some of the data presented are only using 1 siRNA. Addition of a second siRNA is needed to increase rigor and show that the observed effects are not an off-target effect of the siRNA. This is particularly important for Fig. 4 where only 1 siRNA targeting RNaseH2A is used, but these results are critical for interpretation of the study as a whole.
2. Similarly, some of the data are only presented in 1 cell line. For instance, degradation of RNaseH2A in cancer (Fig. 6) is only in HCT116 cells. Addition of a 2nd cell line would increase rigor of the results.
3. One major piece of data that are missing is expression of Lamin B1. For instance, upon overexpression of RNaseH2A in RAS-induced senescent cells, it is possible that the decrease in SASP expression is due to maintenance of Lamin B1 expression. Addition of a western blot or qPCR to show Lamin B1 remains low when RNaseH2A is overexpressed is needed.

Minor Concerns

1. It would be helpful for the authors to include number of independent biological replicates for all experiments either in the figure legend or Methods section. For instance, it looks as though experiments in Fig 3 were only performed once.
2. In Fig S5, p21 is used as a senescence marker. While this could be an indicator of senescence, increased p21 is also associated with other cell phenotypes. This needs to be mentioned.
3. It would be helpful for the authors to comment on why they used TCGA data for only colon, cervical, and ovarian cancer. Is there a biological reason behind this?
4. In the discussion, some of the conclusions need to be slightly tempered to be more in line with the study's results.
 - a. For instance, in the first paragraph "...RNaseH2A dysfunction through downregulation of E2F activity causes accumulation of nucleotide ligands...". This is not actually shown within the manuscript (for instance, that knockdown of E2F1 decreases RNaseH2A and increases cytoplasmic dsDNA).
 - b. Another instance is towards the end of the second paragraph "...SASP factors caused by RNaseH2A ablation might provoke malignant transformation...". The data shown in Fig 6,7 are mostly correlative in this regard, so changing this wording is warranted.
5. A previous report from the Campisi lab found that simply overexpressing p16, which would lead to loss of E2F transcriptional activity, does not promote the SASP. A discussion of how these data and the current study's data can be reconciled is warranted.

Reviewer #2 (Remarks to the Author):

In this article Okada and colleagues very nicely show that RNaseH2A is downregulated during different types of senescence. RNaseH2A is regulated by E2F transcription factors. Depletion of RNaseH2A induces SASP through DNA fragmentation in senescent cells and also in cancer cells. The results are clear, well presented and interesting for the field.

Comments:

1- The authors show that the expression of RNaseH2A is regulated by E2F factors. E2F is inhibited by p16-RB pathway during senescence. Is the downregulation of RNaseH2A dependent on p16-induced senescence? What if p16 is depleted?

2- In the HCT116 cell line, is RNaseH2A also regulated by E2F transcription factors in cancer cells or in cells from the patients with Werner syndrome?

3- The pdf files with supplementary table 1 and table 2 are not well converted. Is not possible to see the name of the gene. Also, there is no Gene name or transcript ID in the early/late data.

Reviewer #3 (Remarks to the Author):

The manuscript entitled „RNaseH2A downregulation drives inflammatory gene expression via genomic DNA fragmentation in senescent and cancer cells“ identifies RNaseH2A as an important regulator of the senescence-associated secretory phenotype (SASP). The authors demonstrate that residual ribonucleoside monophosphates (rNMPs) accumulate in the genome due to the lack of RNaseH2A activity during senescence. This accumulation was associated with genome fragmentation and SASP induction. Interestingly, the authors show that RNaseH2A expression was reduced in cells derived from patients with Werner syndrome but also in aged tissue. Importantly, the SASP gene expression was significantly reduced when RNaseH2A was overexpressed in HRAS-V12-induced senescent cells.

The results are novel and interesting regarding the genome instability during senescence and SASP induction. Moreover, the experiments were performed in human cells and disease relevant cell models. Thus, it is worth communicating these findings to the scientific community. However, prior to publication in Communications Biology few experiments should be performed.

Major concerns:

Figure 1:

The association between RNaseH2A downregulation and SASP induction was observed in models for senescence that involve DNA damage and oncogene activation. It would be important to see whether this association can be found also in models that are not directly linked to DNA damage, such as CDK4/6 inhibitor-mediated senescence. RNaseH2A downregulation and SASP induction should be examined in Abemaciclib and Palbociclib-treated cells.

Figure 2:

The results that the RNaseH2A expression is regulated by E2F transcription factors in non-senescent cells are convincing. However, it is not clear whether E2F activity can lead to increased expression of RNaseH2A in senescent cells. The authors should therefore test the effect of E2F overexpression on RNaseH2A and SASP expression in senescent cells.

Figure 3/Supplementary Figure S3:

Interestingly, the DNA fragmentation was not detected in quiescent TIG-3 cells despite of the downregulation of RNAaseH2A. To support the data, the DNA fragmentation should be examined in other human primary senescent and quiescent cell types.

Figure 4:

The authors claim that downregulation of RNaseH2A by transfection of one siRNA against RNaseH2A results in senescence in TIG3 cells. This observation could be due to an off-target effect of the siRNA. Thus, to strengthen this finding, it is important to test whether the observed senescence phenotype is reproducible using at least two different siRNAs against RNaseH2A. In addition, the question rises whether this senescence phenotype can be found in human primary cells, such as WI-38 or IMR90. Further questions remain regarding the senescence phenotype:
Do these cells stain positive for SA- β GAL activity and do these cells show downregulation of Lamin B1?
Do these cells still proliferate?

It is interesting that the overexpression of RNaseH2A in RAS-induced senescent cells results in a significant decrease in SASP gene expression. Previously, it was shown that cytosolic chromatin fragments and micronuclei accumulate in the cytosol due to the downregulation of Lamin B1. The accumulation of these DNA substrates was associated with the induction of SASP. In this context, it would be interesting to test whether RNaseH2A downregulation results in an increase in cytosolic chromatin fragments. Furthermore, the amount of CCFs should be assessed when overexpressing RNaseH2A in senescent cells.

Point-by-point responses to the reviewers' comments

We thank the three reviewers for their valuable comments and constructive suggestions. We are pleased that they found our results “interesting,” and we have tried to address all of the issues and concerns that they identified.

Reviewers' comments:

Reviewer #1 (Remarks to the Author):

This manuscript by Okada et al. describes a new mechanism whereby the senescence-associated secretory phenotype (SASP) is regulated. This is a highly important question as the SASP has major detrimental effects in both cancer and age-related pathologies. They find that RNaseH2A is decreased during cellular senescence and physiological and pathological aging. This was associated with increased genomic rNMP accumulation and cytoplasmic dsDNA, a known activator of the SASP. Using experiments to knockdown or degrade RNaseH2A, they also observe cytoplasmic dsDNA and SASP expression. This is an intriguing study that provides new insight into the upstream mediators of SASP activation.

Response-1:

We thank you for these thoughtful comments.

Major Concerns

1. Some of the data presented are only using 1 siRNA. Addition of a second siRNA is needed to increase rigor and show that the observed effects are not an off-target effect of the siRNA. This is particularly important for Fig. 4 where only 1 siRNA targeting RNaseH2A is used, but these results are critical for interpretation of the study as a whole.

Response-2:

We completely agree with this comment. In line with your suggestion, we substantially repeated all experiments using multiple siRNAs targeting RNaseH2A and revised the data (see new Fig. 4 and Supplementary Fig. 4). Knockdown of RNaseH2A by two siRNAs in pre-senescent fibroblasts led to phenotypes consistent with our previous data.

2. Similarly, some of the data are only presented in 1 cell line. For instance, degradation of RNaseH2A in cancer (Fig. 6) is only in HCT116 cells. Addition of a 2nd cell line would increase rigor of the results.

Response-3:

We thank you for this constructive suggestion. To further strengthen our data, we have repeated key experiments using another human fibroblast line (IMR-90 cells) (see new Supplementary Figs. 1c, 3b, and 4). Consequently, we confirmed the reproducibility of our data. In addition, we used another cancer cell line in new Supplementary Fig. 8. Coincident with the data using colorectal cancer cells (HCT116), we demonstrated that RNaseH2A knockdown also induced SASP-like gene expression and increased the invasiveness of ovarian cancer cells (SK-OV-3).

3. One major piece of data that are missing is expression of Lamin B1. For instance, upon overexpression of RNaseH2A in RAS-induced senescent cells, it is possible that the decrease in SASP expression is due to maintenance of Lamin B1 expression. Addition of a western blot or qPCR to show Lamin B1 remains low when RNaseH2A is overexpressed is needed.

Response-4:

We appreciate this constructive suggestion. As you suggested, we checked the expression of lamin B1 by RT-qPCR (see new Figs. 4e and 5d and Supplementary Fig. 4d). According to the new Fig. 5d, RNaseH2A overexpression did not rescue lamin B1 expression in HRasV12-induced senescent cells, indicating that the decrease in SASP factor expression induced by RNaseH2A overexpression is not attributable to the maintenance of lamin B1 expression.

Minor Concerns

1. It would be helpful for the authors to include number of independent biological replicates for all experiments either in the figure legend or Methods section. For instance, it looks as though experiments in Fig 3 were only performed once.

Response-5:

We agree with your remarks. In accordance with your suggestion, we have added a description of the experimental replicates in the legends of all figures. In Fig. 3, a gel electrophoresis experiment was reproduced with biological replicates (N = 3), but estimation of the DNA fragment size by mathematical modeling was performed using only the representative data.

2. In Fig S5, p21 is used as a senescence marker. While this could be an indicator of senescence, increased p21 is also associated with other cell phenotypes. This needs to be mentioned.

Response-6:

We agree with this comment. In accordance with your suggestion, we cited some papers that used p21 as a senescence marker (Krizhanovsky *et al.* Cell, 2008; Kang *et al.* Nature, 2011; Takahashi *et al.* Nat. Commun., 2018; Wakita *et al.* Nat. Commun., 2020; Igarashi *et al.* Nat. Commun., 2022). We toned down and rephrased “senescent cells” as “p21-positive cells” in the revised text on page 10, lines 221–223.

3. It would be helpful for the authors to comment on why they used TCGA data for only colon, cervical, and ovarian cancer. Is there a biological reason behind this?

Response-7:

According to the previous report illustrating that RNase H2 functions as a tumor suppressor in colorectal cancers (Aden *et al.*, Gastroenterology, 2019), we checked the association between RNASEH2A expression and the survival rate of patients with colon cancer used TCGA data. In addition, we analyzed the data of patients with other cancers and found a positive correlation between E2F1 and RNaseH2A expression and an inverse correlation between RNaseH2A expression and poor prognosis in cervical and ovarian cancer. However, we do not know the biological meaning of the presence of these phenomena in the three examined types of cancers. Therefore, we mentioned this point in the revised manuscript on page 13, lines 301 to page 14, lines 307.

4. In the discussion, some of the conclusions need to be slightly tempered to be more in line with the study's results.

a. For instance, in the first paragraph “...RNaseH2A dysfunction through downregulation of E2F activity causes accumulation of nucleotide ligands...”. This is not actually shown within the manuscript (for instance, that knockdown of E2F1 decreases RNaseH2A and increases cytoplasmic dsDNA).

b. Another instance is towards the end of the second paragraph “...SASP factors caused by RNaseH2A ablation might provoke malignant transformation...”. The data shown in Fig 6,7 are mostly correlative in this regard, so changing this wording is warranted.

Response-8:

We appreciate your helpful comment. We completely agree with your point, and we have rephrased the sentences in the revised manuscript on page 13, lines 287–291 and page 13, line 301 to page 14, line 307.

5. A previous report from the Campisi lab found that simply overexpressing p16, which would lead to loss of E2F transcriptional activity, does not promote the SASP. A discussion of how these data and the current study's data can be reconciled is warranted.

Response-9:

We thank you for these thoughtful comments. In response to these comments, we conducted RT-qPCR analysis after p16 overexpression. Consistent with the previous data from Dr. Campisi's group, p16 overexpression itself did not induce SASP factor gene expression for 14 days, but it decreased RNASEH2A expression and cell proliferation was inhibited (see the provided figure). These data indicated that RNaseH2A is transcriptionally regulated by E2Fs; therefore, p16 overexpression decreased RNaseH2A expression by downregulating E2F activity. According to our data in this study, RNaseH2A downregulation causes the accumulation of DNA fragments in the cytoplasm of arrested cells, leading to activation of the DNA sensing pathway and upregulation of SASP factor expression (see new Figure 4d, e and Supplementary Fig. 4). Consistent with our findings, several SASP factors are induced in p16-overexpressed cells after long-term incubation (see the provided figure).

RT-qPCR analysis of the indicated genes was conducted using RNA extracted from TIG-3 cells infected with retrovirus encoding p16 or empty vector for 14 days (A) or 28 days (B). Error bars indicate the mean \pm SD of technical triplicates. Two-tailed Student's *t*-test.

Reviewer #2 (Remarks to the Author):

In this article Okada and colleagues very nicely show that RNaseH2A is downregulated during different types of senescence. RNaseH2A is regulated by E2F transcription factors. Depletion of RNaseH2A induces SASP through DNA fragmentation in senescent cells and also in cancer cells. The results are clear, well presented and interesting for the field.

Response-1:

We appreciate your comments and your review of your manuscript.

Comments:

1- The authors show that the expression of RNaseH2A is regulated by E2F factors. E2F is inhibited by p16-RB pathway during senescence. Is the downregulation of RNaseH2A dependent on p16-induced senescence? What if p16 is depleted?

Response-2:

We appreciate your constructive suggestion. We performed p16 (CDKN2A) knockdown experiments using replicative senescent cells. According to the RT-qPCR data, p16 knockdown partially but significantly rescued RNASEH2A mRNA expression (see the provided figure), indicating that RNaseH2A downregulation is dependent on the p16–RB pathway. Additionally, we performed RT-qPCR in p16-overexpressing cells (see Response-9 to Reviewer #1). We found that p16 overexpression itself decreased RNASEH2A mRNA expression, indicating that RNaseH2A expression is dependent on the p16–RB pathway.

RT-qPCR analysis of the indicated genes was conducted using RNA extracted from pre-senescent or replicatively senescent (RS) TIG-3 cells treated with siRNA against CDKN2A. One-way ANOVA coupled with Dunnett's multiple comparisons test.

2- In the HCT116 cell line, is RNaseH2A also regulated by E2F transcription factors in cancer cells or in cells from the patients with Werner syndrome?

Response-3:

We appreciate your helpful comment. To answer this question, we performed DP1 knockdown experiments in cells (WF1A) taken from a patient with Werner syndrome. Knockdown of DP1 significantly decreased RNaseH2A expression at both the mRNA and protein levels in WF1A cells (see new Supplementary Figs. 7h and i), indicating that RNaseH2A expression is regulated by E2F transcription factors in cells from patients with Werner syndrome.

Conversely, DP1 knockdown in HCT116 cells did not repress RNaseH2A expression. Because we speculated that DP1 knockdown is insufficient to block E2F activity in HCT116 cells, we conducted CDK4/6 inhibitor treatment using palbociclib and abemaciclib in HCT116 cells. We found that RNaseH2A expression was significantly downregulated by treatment with both CDK4/6 inhibitors (see the provided figure). These results suggested that RNaseH2A expression is regulated by E2F transcription factors in cancer cells.

RT-qPCR analysis of RNaseH2a mRNA expression in HCT116 cells treated with CDK4/6 inhibitors. Error bars indicate the mean \pm SD of technical triplicates. One-way ANOVA coupled with Dunnett's multiple comparisons test.

3- The pdf files with supplementary table 1 and table 2 are not well converted. Is not possible to see the name of the gene. Also, there is no Gene name or transcript ID in the early/late data.

Response-4:

We deeply apologize for our oversight. In accordance with your suggestion, we have carefully revised Supplementary Tables 1–2 to be able to ensure that the gene names are visible.

Reviewer #3 (Remarks to the Author):

The manuscript entitled „RNaseH2A downregulation drives inflammatory gene expression via genomic DNA fragmentation in senescent and cancer cells” identifies RNaseH2A as an important regulator of the senescence-associated secretory phenotype (SASP). The authors demonstrate that residual ribonucleoside monophosphates (rNMPs) accumulate in the genome due to the lack of RNaseH2A activity during senescence. This accumulation was associated with genome fragmentation and SASP induction. Interestingly, the authors show that RNaseH2A expression was reduced in cells derived from patients with Werner syndrome but also in aged tissue. Importantly, the SASP gene expression was significantly reduced when RNaseH2A was overexpressed in HRAS-V12-induced senescent cells.

The results are novel and interesting regarding the genome instability during senescence and SASP induction. Moreover, the experiments were performed in human cells and disease relevant cell models. Thus, it is worth communicating these findings to the scientific community. However, prior to publication in Communications Biology few experiments should be performed.

Response-1:

We appreciate your fruitful comments.

Major concerns:

Figure 1:

The association between RNaseH2A downregulation and SASP induction was observed in models for senescence that involve DNA damage and oncogene activation. It would be important to see whether this association can be found also in models that are not directly linked to DNA damage, such as CDK4/6 inhibitor-mediated senescence. RNaseH2A downregulation and SASP induction should be examined in Abemaciclib and Palbociclib-treated cells.

Response-2:

We thank you for these helpful comments. In accordance with your remarks, we treated pre-senescent fibroblasts with CDK4/6 inhibitors for 7 days. RT-qPCR revealed that CDK4/6 inhibitor treatment significantly reduced RNASEH2A expression (see new

Supplementary Fig. 2f). These data strongly indicated that RNaseH2A expression is regulated by E2F transcription factors.

Figure 2:

The results that the RNaseH2A expression is regulated by E2F transcription factors in non-senescent cells are convincing. However, it is not clear whether E2F activity can lead to increased expression of RNaseH2A in senescent cells. The authors should therefore test the effect of E2F overexpression on RNaseH2A and SASP expression in senescent cells.

Response-3:

In accordance with your comment, we overexpressed E2F3 in X-ray-induced senescent IMR-90 cells and found that E2F3 overexpression did not increase the expression of RNASEH2A in senescent IMR-90 cells (see the provided figure). Generally, E2F activity is strongly suppressed by unphosphorylated (activated) RB protein in senescent cells (Gorgoulis *et al.*, Cell, 2019), and the loci of E2F target genes were sequestered to heterochromatin and silenced in senescent cells, as reported by Dr. Lowe's group (Narita *et al.*, Cell, 2003). Therefore, it might also be regulated by epigenetic mechanisms, and E2F overexpression alone is insufficient for recovering RNASEH2A expression in senescent cells, at least in our experimental settings. Meanwhile, p16 knockdown partially but significantly rescued RNASEH2A mRNA expression in replicative senescent cells (see Response-2 to Reviewer #2). We believe that RNaseH2A expression is partially regulated by E2F transcription factors in senescent cells.

RT-qPCR analysis of the indicated genes was conducted using RNA extracted from proliferating or X-ray-induced senescent IMR-90 cells infected with lentivirus encoding E2F3 or GFP (negative control). Error bars indicate the mean \pm SD of technical triplicates. Two-way ANOVA coupled with Tukey's multiple comparisons test.

Figure 3/Supplementary Figure S3:

Interestingly, the DNA fragmentation was not detected in quiescent TIG-3 cells despite of the downregulation of RNAaseH2A. To support the data, the DNA fragmentation should be examined in other human primary senescent and quiescent cell types.

Response-4:

We thank you for this advice. To strengthen our findings, we have repeated the experiments of agarose gel electrophoresis using genomic DNA and added data using IMR-90 cells. The mobility of genomic DNA from senescent IMR-90 cells tended to increase following alkaline treatment compared to that from pre-senescent IMR-90 cells, whereas the mobility of genomic DNA from quiescent IMR-90 was not significantly changed (see new Supplementary Fig. 3b and the provided figure). These data indicated that genomic rNMPs are accumulated in senescent, but not in quiescent, IMR-90 cells.

IMR-90 cells were rendered quiescent by serum starvation and subjected to genome DNA extraction. Genome DNA (3 µg) was subjected to agarose gel electrophoresis with or without alkaline hydrolysis (see also Supplementary Fig. S3b).

Figure 4:

The authors claim that downregulation of RNaseH2A by transfection of one siRNA against RNaseH2A results in senescence in TIG3 cells. This observation could be due to an off-target effect of the siRNA. Thus, to strengthen this finding, it is important to test whether the observed senescence phenotype is reproducible using at least two different siRNAs against RNaseH2A. In addition, the question rises whether this senescence phenotype can be found in human primary cells, such as WI-38 or IMR90. Further questions remain regarding the senescence phenotype: Do these cells stain positive for SA-βGAL activity and do these cells show downregulation of Lamin B1? Do these cells still proliferate?

Response-5:

We are grateful for such fruitful comments, and we agree with your remarks. In line with your comments, we substantially revised the data using multiple siRNAs targeting RNASEH2A in TIG-3 and IMR-90 cells (see new Fig. 4 and Supplementary Fig. 4). Therefore, we concluded that these senescence phenotypes induced by RNaseH2A knockdown were reproducible, and they were also observed in another primary fibroblast line (IMR-90 cells). In addition, we performed SA- β -GAL staining in TIG-3 and IMR-90 cells and found that RNaseH2A knockdown significantly increased the number of SA- β -GAL-positive cells, which were not proliferative (see new Fig. 4b and Supplementary Fig. 4b). Furthermore, we confirmed that lamin B1 expression was significantly decreased by RNaseH2A knockdown in both TIG-3 and IMR-90 cells (see new Fig. 4e and Supplementary Fig. 4d).

It is interesting that the overexpression of RNaseH2A in RAS-induced senescent cells results in a significant decrease in SASP gene expression. Previously, it was shown that cytosolic chromatin fragments and micronuclei accumulate in the cytosol due to the downregulation of Lamin B1. The accumulation of these DNA substrates was associated with the induction of SASP. In this context, it would be interesting to test whether RNaseH2A downregulation results in an increase in cytosolic chromatin fragments. Furthermore, the amount of CCFs should be assessed when overexpressing RNaseH2A in senescent cells.

Response-6:

We thank you for your helpful comments. In accordance with your suggestion, we checked the amounts of cytoplasmic DNA fragments (CCFs) and micronuclei-positive cells in RNaseH2A-overexpressing senescent cells. Although the amounts of CCFs were significantly decreased (see new Fig. 5c), the percentage of micronuclei-positive cells was not changed by RNaseH2A overexpression in HRasV12-induced senescent cells (the provided figure). These data indicate that RNaseH2A is important for the repression of CCF accumulation but not for micronuclei formation, differing from lamin B1. RNaseH2A overexpression clearly blocked the expression of SASP genes in HRasV12-induced senescent cells (see new Fig. 5d). Therefore, we consider that CCFs rather than micronuclei are critical for SASP factor gene expression in HRasV12-induced senescent cells, at least in our experimental settings (see new Fig. 5d).

A TIG-3 cells infected with retrovirus encoding flag-tagged RNaseH2A or empty vector. After puromycin selection, cells were rendered senescent by treatment with HRasV12-encoding retrovirus and subjected to immunofluorescence. dsDNA (green), lamin B1 (red), and DAPI (blue). White arrows denote dsDNA-positive micronuclei. Scale bars, 20 μ m.

B Quantification of micronuclei-positive cells. Error bars indicate the mean \pm SD of three biological replicates. Two-way ANOVA coupled with Tukey's multiple comparisons test.

REVIEWERS' COMMENTS:

Reviewer #1 (Remarks to the Author):

The authors have substantially addressed my concerns. Overall, this is a very elegant study with clear implications for the senescence field.

Reviewer #2 (Remarks to the Author):

In this new version of the manuscript the authors have answered my concerns. Together with the answer to the other reviewers, the revised manuscript has been improved. I recommend the manuscript for publication.

Reviewer #3 (Remarks to the Author):

The manuscript by Okada et al. identifies RNaseH2A as a mediator and regulator of the senescence-associated secretory phenotype (SASP). This finding is novel and interesting for the field. The authors addressed all the comments by the reviewers. The results support the conclusions.